# Mutual modulation of gut microbiota and the immune system in type 1 diabetes models

Estela Rosell-Mases [1], Alba Santiago[2], Marta Corral-Pujol[1], Francisca Yáñez[2], Encarna Varela[2], Leire Egia-Mendikute[1], Berta Arpa[1], Catalina Cosovanu [1], Anaïs Panosa [3], Gerard Serrano-Gómez [2], Conchi Mora[1], Joan Verdaguer [1,4,6] ✉ & Chaysavanh Manichanh [2,5,6] ✉

The transgenic 116C-NOD mouse strain exhibits a prevalent Th17 phenotype, and reduced type 1 diabetes (T1D) compared to non-obese diabetic (NOD) mice. A cohousing experiment between both models revealed lower T1D incidence in NOD mice cohoused with 116C-NOD, associated with gut microbiota changes, reduced intestinal permeability, shifts in T and B cell subsets, and a transition from Th1 to Th17 responses. Distinct gut bacterial signatures were linked to T1D in each group. Using a RAG-2$^{-/-}$ genetic background, we found that T cell alterations promoted segmented filamentous bacteria proliferation in young NOD and 116C-NOD, as well as in immunodeficient NOD.RAG-2$^{-/-}$ and 116C-NOD.RAG-2$^{-/-}$ mice across all ages. *Bifidobacterium* colonization depended on lymphocytes and thrived in a non-diabetogenic environment. Additionally, 116C-NOD B cells in 116C-NOD.RAG-2$^{-/-}$ mice enriched the gut microbiota in *Adlercreutzia* and reduced intestinal permeability. Collectively, these results indicate reciprocal modulation between gut microbiota and the immune system in rodent T1D models.

Type 1 diabetes (T1D) is a chronic autoimmune disease characterised by the destruction of insulin-producing β-cells of the pancreas[1]. The diabetic autoimmune process involves the infiltration of Langerhans islets by immune system cells, a phenomenon known as insulitis[2]. Currently, one of the most used animal models to investigate this disease is the non-obese diabetic (NOD) mouse, which spontaneously develops autoimmune diabetes in a form that, in many aspects, resembles the human disease[3].

T1D occurs as a consequence of a strong Th1 response[4,5], in which both CD4$^+$ and CD8$^+$ T lymphocytes are the major effectors of β-cell damage in humans[6] and NOD mice[7,8]. However, B cells are also essential for the pathogenesis of T1D in humans[9] and the NOD murine model[10]. To thoroughly analyse the relevance of B lymphocytes in T1D,

our team previously generated the 116C-NOD transgenic mouse, which carries the immunoglobulin genes of islet-infiltrating and β-cell reactive B lymphocytes from a diabetes-resistant but insulitis-prone mouse. Despite their anergic-like phenotype, 116C-NOD B cells induce a T cell shift towards a Th17 phenotype, which results in decreased incidence of diabetes in the 116C-NOD compared with NOD mice[11].

Since T and B cells are the main mediators of T1D, NOD mice lacking both lymphocyte populations are a powerful tool to analyse adaptive immunity in T1D. In this regard, the RAG-2 (recombination activating gene 2)-deficient NOD (NOD.RAG-2$^{-/-}$) which does not develop T1D, was engineered[8]. Going further and taking advantage of both 116C-NOD and NOD.RAG-2$^{-/-}$ models, we generated the RAG-2

[1]Immunology and Immunopathology Group, Department of Experimental Medicine, Faculty of Medicine, Universitat de Lleida (UdL) and Institut de Recerca Biomèdica de Lleida (IRBLleida), 25198 Lleida, Spain. [2]Microbiome Lab, Vall d'Hebron Institut de Recerca (VHIR), Vall d'Hebron Barcelona Hospital Campus, 08035 Barcelona, Spain. [3]Flow Cytometry Facility, Universitat de Lleida (UdL) and Institut de Recerca Biomèdica de Lleida (IRBLleida), 25198 Lleida, Spain. [4]CIBER of Diabetes and Associated Metabolic Diseases (CIBERDEM), Instituto de Salud Carlos III (ISCIII), 28029 Madrid, Spain. [5]CIBER of Hepatic and Digestive Diseases (CIBEREHD), Instituto de Salud Carlos III (ISCIII), 28029 Madrid, Spain. [6]These authors contributed equally: Joan Verdaguer, Chaysavanh Manichanh. ✉e-mail: joan.verdaguer@udl.cat; cmanicha@gmail.com

deficient 116C-NOD (116C-NOD.RAG-2[-/-]) mouse[11] to exclusively study the role of 116C B lymphocytes. Despite their intact B cell compartment, 116C-NOD.RAG2[-/-] mice do not develop diabetes due to a lack of T cells.

Over the last 20 years, several clinical studies found an association between T1D and gut microbiota composition[12,13]. Experiments with animal models have gone one step further than mere association, demonstrating that T1D can be promoted or inhibited through the modulation of gut microbiota[14]. Nevertheless, few studies have focused their attention on the link between intestinal microbiota and the immune system, both innate[15–17] and adaptive[18–21], in the context of T1D.

Here, we present an experimental study on the cross-talk between the gut microbiota and the immune system in the T1D process, using the NOD mouse and transgenic and knockout variants. First, to evaluate the modulation of B and T cell response, and thereby T1D development by the gut microbiota, NOD mice were cohoused with the 116C-NOD B-cell transgenic model. The cytokine secretion profile, the major CD4[+] Th response transcription factors, the T and B cell subsets, as well as the gut microbiota and intestinal permeability of NOD and 116C-NOD mice, were analysed. Second, to further explore

the influence of the adaptive immune system of NOD and 116C-NOD models on their faecal microbiota, the gut microbiota and intestinal wall integrity of the immunodeficient variants NOD.RAG-2[-/-] and 116C-NOD.RAG-2[-/-], as well as a non-T1D-prone mouse control, were also studied. Lastly, to assess the effect of T and B cells on the composition of the intestinal bacterial populations, lymphocyte transfers were performed into immunodeficient recipients, and differences in gut microbiota composition post-transfer were analysed.

## Results

### NOD and 116C-NOD mice cohousing reduces and increases their incidence and hazard rate of T1D, respectively

116C-NOD transgenic mice developed a significantly lower cumulative incidence of T1D ($p = 0.007$), with half the diabetes occurrence rate, compared with wild-type NOD mice (HR = 0.489) (Fig. 1a). To evaluate the effect of gut microbiota from both strains on the development of T1D, NOD and 116C-NOD littermates were housed in isolation or cohabitation conditions.

Disease monitoring revealed that NOD cohoused with 116C-NOD showed a significantly diminished incidence of T1D ($p = 0.010$), with

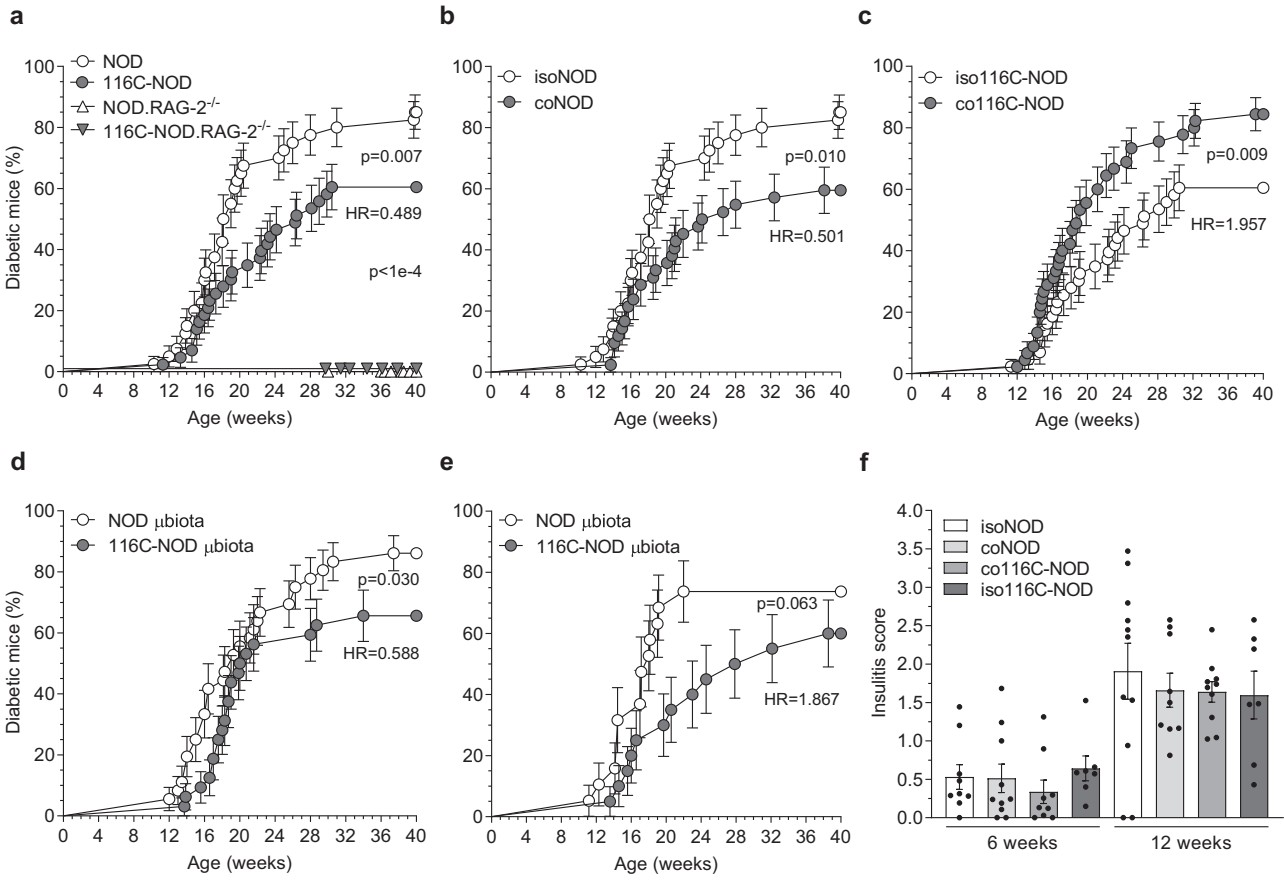

**Fig. 1 | T1D monitoring of NOD and 116C-NOD mouse models under natural transfer of gut microbiota. a** Diabetes incidence of NOD ($n = 40$), 116C-NOD ($n = 43$), NOD.RAG-2[-/-] ($n = 17$) and 116C-NOD.RAG-2[-/-] ($n = 23$) mice. The four models were housed separately. **b** Diabetes incidence of NOD mice isolated (iso-NOD) ($n = 40$) and cohoused (coNOD) ($n = 42$) with their 116C-NOD littermates. **c** Diabetes incidence of 116C-NOD mice isolated (iso116C-NOD) ($n = 43$) and cohoused (co116C-NOD) ($n = 45$) with their NOD littermates. **d** Diabetes incidence of NOD mice recipients of microbiota from NOD (NOD μbiota) ($n = 36$) and 116C-NOD (116C-NOD μbiota) ($n = 32$). **e** Diabetes incidence of 116C-NOD mice recipients of microbiota from NOD (NOD μbiota) ($n = 19$) and 116C-NOD (116C-NOD μbiota) ($n = 20$). Microbiota recipients were housed in cages previously occupied by NOD or 116C-NOD mice. Cage change was performed thrice a week. Diabetes incidence

curves are expressed as mean ± SE and analysed with the Log-rank (Mantel-Cox) test. Two-tailed $p$-values were obtained for (**a**), (**b**), and (**c**) data, as well as one-tailed $p$-values were considered for (**d**) and (**e**) data. Hazard ratios (HR) of 116C-NOD/NOD in (**a**), coNOD/isoNOD in (**b**), co116C-NOD/iso116C-NOD in (**c**), 116C-NOD μbiota/NOD μbiota in (**d**) and NOD μbiota/116C-NOD μbiota in (**e**) were analysed with Mantel-Haenszel test's. **f** Insulitis score of isolated NOD (isoNOD) (6 weeks: $n = 9$, 12 weeks: $n = 11$), cohoused (coNOD) (6 weeks: $n = 10$, 12 weeks: $n = 9$), cohoused 116C-NOD (co116C-NOD) (6 weeks: $n = 9$, 12 weeks: $n = 10$), and isolated 116C-NOD (iso116C-NOD) (6 weeks: $n = 7$, 12 weeks: $n = 7$). Insulitis data are expressed as mean ± SE and analysed with Mann–Whitney's test (one-sided, no significant differences were found).

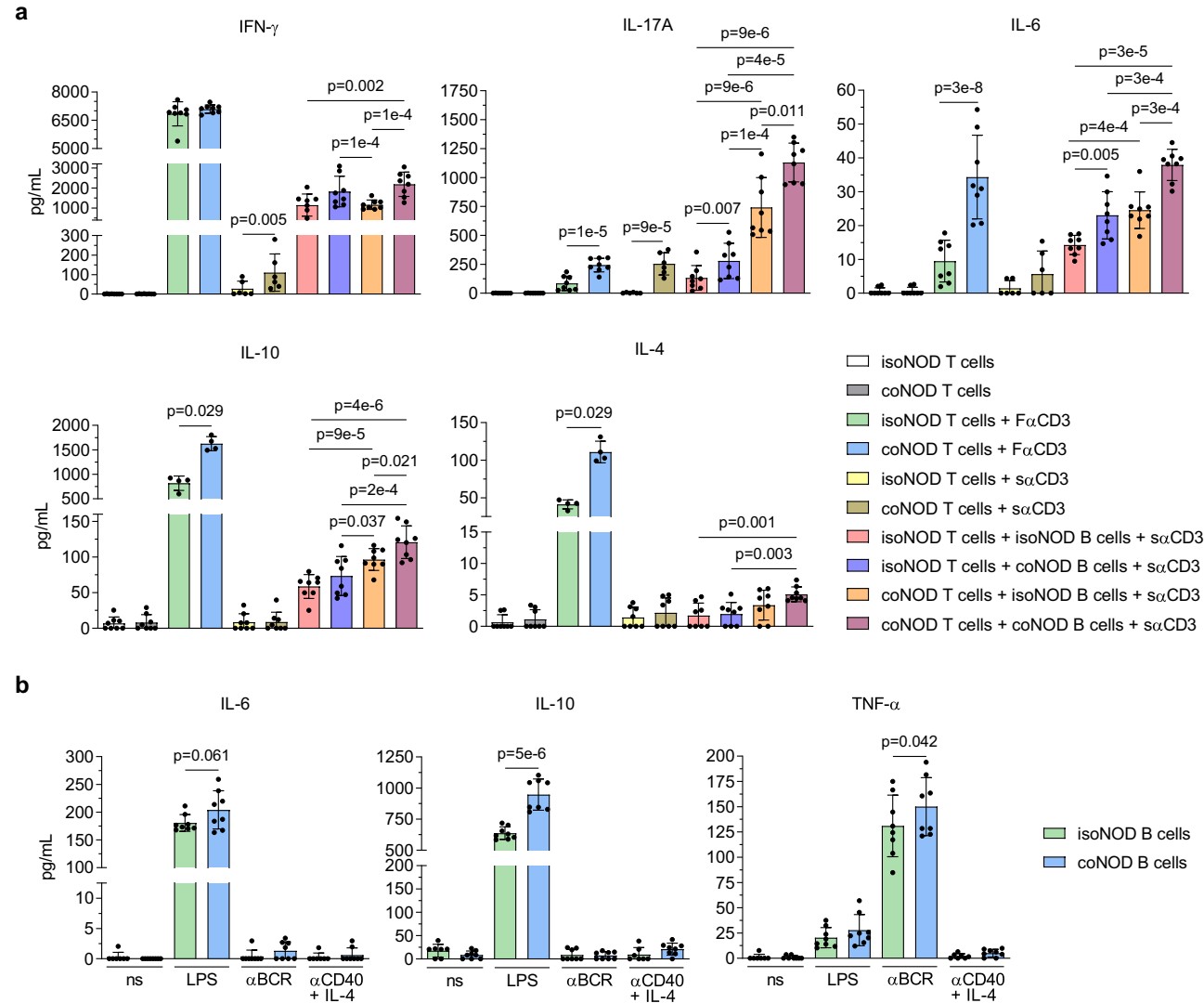

**Fig. 2 | In vitro cytokine secretion analysis of T and B lymphocytes from NOD mice isolated and cohoused with 116C-NOD mice. a** T cells from NOD mice isolated (isoNOD T cells) and cohoused (coNOD T cells) were cultured in vitro under different conditions: alone (isoNOD and coNOD: $n = 8$), with well-coated or fixed anti-CD3 (FαCD3) (isoNOD and coNOD: $n = 8$ for IFN-γ, IL-17A, and IL-6; $n = 4$ for IL-10 and IL-4), in the presence of soluble anti-CD3 (sαCD3) (isoNOD and coNOD: $n = 6$ for IFN-γ, IL-17A, and IL-6; $n = 8$ for IL-10 and IL-4), and co-cultured with B cells from NOD mice isolated (isoNOD B cells) and cohoused (coNOD B cells), in their four possible combinations, plus sαCD3: isoNOD T cells + isoNOD B cells ($n = 8$ for all cytokines), isoNOD T cells + coNOD B cells ($n = 8$ for all cytokines),

coNOD T cells + isoNOD B cells ($n = 8$ for all cytokines), and coNOD T cells + coNOD B cells ($n = 8$ for all cytokines). **b** B cells from NOD mice isolated (isoNOD B cells) and cohoused (coNOD B cells) were cultured under different conditions: without stimulus (ns) (isoNOD: $n = 7$, coNOD: $n = 8$), with lipopolysaccharide (LPS) (isoNOD and coNOD: $n = 8$), with anti-B cell receptor (αBCR) (isoNOD and coNOD: $n = 8$), and with anti-CD40 (αCD40) plus IL-4 (isoNOD: $n = 7$, coNOD: $n = 8$). Two independent experiments were conducted (both shown). Data are expressed as mean ± SD and analysed with two-way ANOVA test (two-sided) on rank-transformed data-values.

half the diabetes occurrence rate (HR = 0.501), compared with NOD housed separately from 116C-NOD mice (Fig. 1b). Conversely, 116C-NOD cohoused with NOD displayed significantly increased incidence of T1D ($p = 0.009$), with twice the diabetes occurrence rate (HR = 1.957), compared with 116C-NOD isolated from NOD mice (Fig. 1c).

In a subsequent experiment, NOD housed in cages previously inhabited by 116C-NOD mice (thus being recipients of microbiota from 116C-NOD) unveiled lower T1D incidence ($p = 0.030$), with a reduced diabetes occurrence rate (HR = 0.588), than those housed in cages previously occupied by other NOD mice (Fig. 1d). Inversely, 116C-NOD mice housed in cages occupied before by NOD mice exhibited increased T1D incidence ($p = 0.063$) and a higher diabetes occurrence rate (HR = 1.867) than those housed in cages occupied before by other 116C-NOD mice (Fig. 1e). Thereby, the cage change experiment, although performed only thrice a week, confirmed the previous results.

No differences were observed in the degree of islet infiltration (insulitis score) between cohoused and isolated mice. However, cohoused NOD exhibited a downward trend compared with isolated NOD mice (Fig. 1f).

## 116C-NOD gut microbiota modulates the cytokine secretion pattern of NOD T and B cells

The cohoused and isolated NOD mice were selected to elucidate the immunological status behind the aforementioned changes in T1D development. In vitro T and B cell cytokine production from both mice was analysed (Fig. 2).

Results indicated that T cells derived from cohoused NOD (coNOD T cells), co-cultured with B cells from cohoused NOD mice (coNOD B cells), secreted higher amounts of IL-17A, IL-6, IL-10, IL-4, and IFN-γ than T cells from isolated NOD (isoNOD T cells) co-cultured

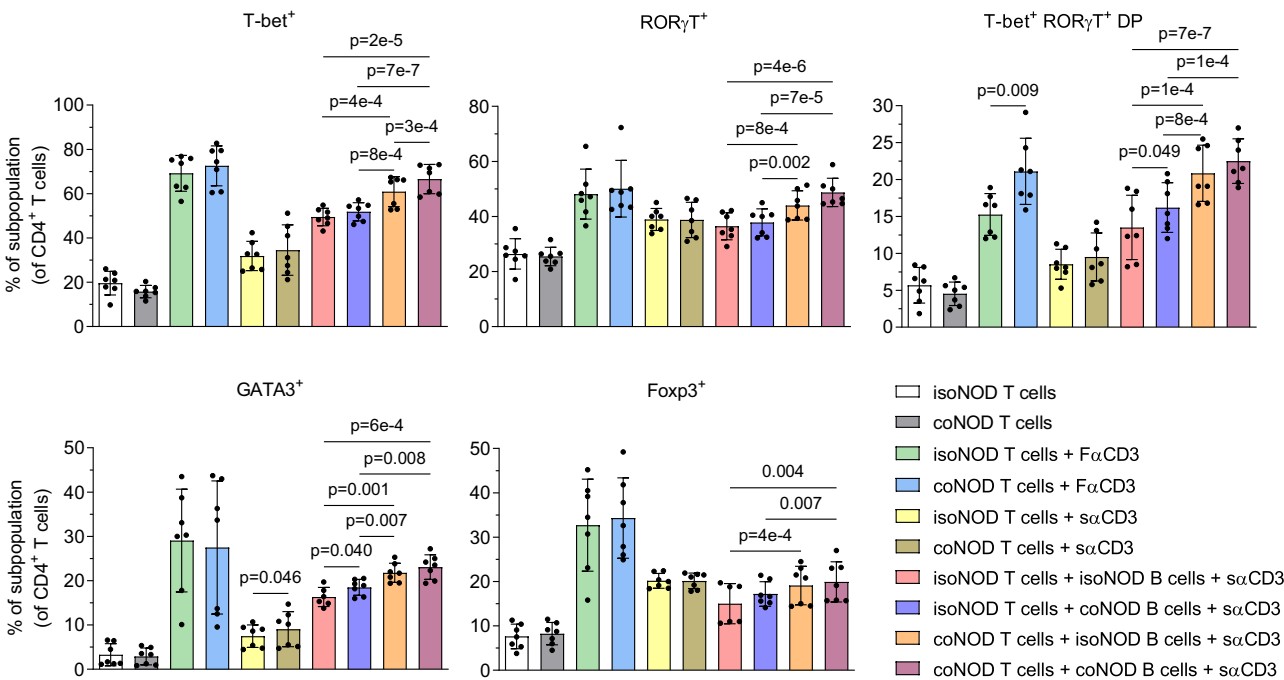

**Fig. 3 | Expression of the major Th transcription factors by in vitro cultured CD4⁺ T lymphocytes from NOD mice isolated and cohoused with 116C-NOD mice.** Expression of the transcription factors T-bet, RORγT, GATA3, and Foxp3 in T cells from NOD mice isolated (isoNOD T cells) and cohoused (coNOD T cells) cultured in vitro under the stimuli previously described in Fig. 2 (*n* = 7 for isoNOD and coNOD T cells in each culture condition). DP = double positive population for the markers in question and negative for the rest of the markers. Two independent experiments were performed (both depicted). Data are expressed as mean ± SD and analysed with two-way ANOVA test (two-sided) on rank-transformed data-values.

with B cells from isolated NOD mice (isoNOD B cells), in the presence of soluble anti-CD3 monoclonal antibody (sαCD3) (Fig. 2a).

The greatest differences were observed in IL-17A secretion, with a mean tenfold increase in coNOD T cells co-cultured with coNOD B cells compared with isoNOD T cells co-cultured with isoNOD B cells. Between these two conditions, a staggering production of IL-17A was found by coNOD T cells co-cultured with isoNOD B cells and isoNOD T cells co-cultured with coNOD B cells. Furthermore, stimulation only with sαCD3 or well-coated anti-CD3 monoclonal antibody (FαCD3) increased the secretion of IL-17A by coNOD T cells. Therefore, coNOD T cells had a much greater capacity for IL-17A secretion, which was enhanced by coNOD B cells.

Unlike IL-17A, in the T and B co-culture conditions, the secretion of IL-6, IL-10, and IL-4 was more balanced between isoNOD T cells plus coNOD B cells and coNOD T cells plus isoNOD B cells, flanked with the highest production by coNOD T cells plus coNOD B cells and the lowest by isoNOD T cells plus isoNOD B cells. Moreover, the secretion of these cytokines was practically absent when T cells were only stimulated with sαCD3, compared with the raised levels in FαCD3 conditions, where coNOD T cells secreted higher amounts than isoNOD T cells. Altogether, these results showed that coNOD B cells were more determinant in inducing the secretion of IL-6, IL-10, and IL-4 by coNOD T cells than in the case of IL-17A.

The pattern of IFN-γ secretion was different from the previous cytokines. IFN-γ production was induced by sαCD3 alone in coNOD T cells, but not in isoNOD T cells. In parallel, under FαCD3 conditions, both coNOD T cells and isoNOD T cells produced high and comparable amounts of this cytokine. Nevertheless, higher levels of IFN-γ were detected when T lymphocytes, either coNOD T cells or isoNOD T cells, were co-cultured with coNOD B cells, compared with isoNOD B cells. Thus, the T cell's ability to secrete IFN-γ was the most influenced by the presence of coNOD B cells in the culture.

Overall, the cytokine secretion analyses of co-cultured T and B cells indicated that T cells from cohoused NOD mice, compared with isolated NOD, had a superior production capacity of Th17, Treg, and Th2 cytokines, positively influenced by B cells from cohoused NOD.

Regarding B lymphocyte culture (Fig. 2b), coNOD B cells produced higher amounts of IL-6 (trend) and IL-10 than isoNOD B cells, in this case, when B lymphocytes were stimulated with the bacterial endotoxin lipopolysaccharide (LPS). In addition, coNOD B cells secreted more TNF-α than isoNOD B cells when stimulated with F(ab')₂ polyclonal anti-IgM (αBCR). Both isoNOD and coNOD B cells did not produce IL-17A, IFN-γ, nor IL-4 (Supplementary Fig. 1).

Supplementary Fig. 2 shows the proliferation index of iso/coNOD T and B cells.

## Expression of the major transcription factors of the Th response is altered in cohoused NOD mice

To further study the response of T and B lymphocytes, the expression levels of the Th-response-defining transcription factors, including T-bet, GATA3, RORγT, and Foxp3 were measured in vitro in cultured T cells from cohoused NOD (coNOD T cells) and isolated NOD (isoNOD T cells) mice (Fig. 3).

The percentage of the CD4⁺ T lymphocyte subpopulations T-bet⁺, RORγT⁺, and T-bet⁺ RORγT⁺ double positive (DP) were significantly higher in coNOD T cells compared with isoNOD T cells, either when co-cultured with coNOD B cells or isoNOD B cells (and in the presence of sαCD3). The most remarkable differences were observed between the co-cultures of coNOD T cells plus coNOD B cells and isoNOD T cells plus isoNOD T cells.

In addition, in the presence of coNOD B cells (compared with co-culture with isoNOD B cells), the percentage of T-bet⁺ significantly increased in coNOD T cells, as well as the percentage of T-bet⁺ RORγT⁺ DP increased in isoNOD T cells. Furthermore, coNOD T cells co-

cultured with isoNOD B cells displayed significantly increased percentages of T-bet$^+$, RORγT$^+$, and T-bet$^+$ RORγT$^+$ DP, compared with isoNOD T cells co-cultured with coNOD B cells.

Lastly, the CD4$^+$ GATA3$^+$ and Foxp3$^+$ T lymphocyte subpopulations showed higher percentages in coNOD T cells compared with isoNOD T cells.

Overall, and consistent with the changes in the cytokine pattern, these findings indicated that T cells of cohoused NOD mice mainly displayed a Th1/Th17 pattern, highly modulated by their B cells.

### Cohousing shapes the T and B cell subsets of mesenteric lymph nodes, gut-associated lymphoid tissue and pancreatic islet infiltrate

With the aim of analysing the immunophenotype of T and B cells directly ex vivo in isolated and cohoused NOD mice, we assessed different lymphocyte subsets in the spleen, mesenteric lymph nodes (MLN), Peyer's patches (PP), caecal patch (CP), and pancreatic mononuclear islet infiltrate (Fig. 4 and Supplementary Fig. 3). The intra-islet infiltrate was analysed to complement the insulitis score study.

Regarding T lymphocytes, higher percentages of effector CD4$^+$ and CD8$^+$ T cells were observed in secondary lymphoid organs, as well as in the islet infiltrate of isolated NOD mice. We also noted increased percentages of exhausted-like PD-1$^+$ CD4$^+$ and CD8$^+$ T cells in these organs of isolated animals. Moreover, anergic-like CD4$^+$ T cells were found in higher percentages in pancreatic islets of cohoused NOD mice (Fig. 4).

In relation to B lymphocytes, while cohoused NOD mice showed an increase of total B cells in PP, CP and pancreatic islets, isolated NOD mice exhibited higher percentages of follicular B cells in secondary lymphoid organs. Additionally, memory B cells were increased in the MLN of isolated animals. Similar to T cells, anergic B cells were found in higher percentages in the islets of cohoused NOD mice. Nevertheless, unlike T cells, anergic B cells displayed higher percentages in the spleen, MLN, PP, and CP of cohoused NOD mice (Fig. 4).

The differences observed in T and B cell subsets within the pancreatic islets indicated that the composition of the islet infiltrate, rather than the degree of infiltration, was influenced by the cohousing of NOD with 116C-NOD mice.

### Each group of diabetes-prone mice possesses a distinct gut bacterial signature

To look into the gut microbiota pattern causing the previously mentioned modulation of the immunological profile and the T1D development, samples from isolated NOD, cohoused NOD, and isolated 116C-NOD mice groups were analysed. The effect of 116C B cell transgenesis and the natural transfer of 116C-NOD intestinal microbiota on the gut bacterial communities of NOD mice were analysed by 16S rRNA gene-sequencing in a longitudinal setting. At six, 12 weeks, and 20 weeks of age, isolated NOD, cohoused NOD, and isolated 116C-NOD mice were divided into two groups: mice that developed diabetes during the 40-week disease follow-up (future diabetics) and mice that remained resistant throughout this period (future-resistant animals). Future-diabetic and future-resistant mice were compared (Fig. 5). The results indicated that specific bacterial taxa could act as markers for T1D in isolated NOD, cohoused NOD, and 116C-NOD mice. Interestingly, 116C transgenesis and cohousing conditions of NOD mice with 116C-NOD led to different bacterial markers for each group of animals, establishing a model-dependent gut microbiota fingerprint for T1D.

First, in the group of isolated NOD mice, at 12 weeks of age, future diabetics exhibited a higher relative abundance of Cyanobacteria, compared with future-resistant animals (false discovery rate (FDR) $q = 0.053$). Second, regarding cohoused NOD mice, the bacterial marker of T1D corresponded to the phylum Proteobacteria, the relative abundance of which was reduced at 12 weeks of age in future-diabetic animals compared with future-resistant ($q = 0.101$). At this

time point, the richness of faecal bacteria, based on Chao1 index analysis, was lower in future diabetics than in future-resistant mice ($p = 0.021$). Third, future-diabetic isolated 116C-NOD mice exhibited an increased relative abundance of the phylum Tenericutes at six weeks of age ($q = 0.049$) compared with future-resistant mice, and also when comparing correlated samples of six, 12, and 20 weeks of future diabetics with future-resistant mice ($q = 0.093$). Furthermore, the unknown genus of Cyanobacteria order YS2 was enriched in correlated samples of six and 12-week-old future-diabetic mice ($q = 0.052$). In parallel, the unknown genus of the Alphaproteobacteria order RF32 was enriched in related samples of six-, 12-, and 20-week-old future-diabetic mice ($q = 0.043$). Finally, as a general T1D bacterial marker for the three groups of diabetes-prone mice, the genus *Clostridium* (from the family Ruminococcaceae) was enriched in the gut microbiota of 20-week-old future diabetics compared with future-resistant animals ($q = 0.094$).

Furthermore, in the same groups of mice, newly diagnosed diabetics were compared to future-resistant animals. This was performed with future-resistant mice at 12 and 20 weeks, but not at six, as 12 weeks was the age at which the mice analysed began to develop T1D (Fig. 6).

The results showed that, once cohoused NOD mice developed T1D, they exhibited a higher relative abundance of an unknown genus of the Mogibacteriaceae family, compared with correlated samples of future-resistant mice at 12 and 20 weeks ($q = 0.080$). Moreover, in isolated 116C-NOD mice, the Cyanobacteria signature of future-diabetic mice was maintained once they became diabetic, as shown when comparing diabetics with future-resistant animals at 12 and 20 weeks of age ($q = 0.032$). Regarding Chao1 index, the richness of the intestinal microbiota in diabetics compared with future-resistant mice also varied depending on the group of animals. In cohoused NOD mice, richness was lower in diabetics than in 12-week-old future-resistant animals ($p = 0.042$). However, neither diabetic mice from the isolated NOD group nor 116C-NOD exhibited a significant difference in comparison to future-resistant animals at 12 or 20 weeks.

In a subsequent analysis, we selected significant bacterial taxa and Chao1 index and grouped future-diabetic and future-resistant T1D-prone mice (isolated NOD, cohoused NOD, and isolated 116C-NOD) based on high or low relative abundance of corresponding taxa or richness. T1D incidence was compared (Supplementary Fig. 4). Isolated NOD mice at 12 weeks with low Cyanobacteria abundance had lower future-diabetes incidence than those with high abundance ($p = 0.039$). At the same age, cohoused NOD mice with high richness (Chao1 index) showed significantly reduced future-T1D incidence compared to those with low richness ($p = 8 \times 10^{-4}$). Additionally, at 12 weeks, cohoused NOD mice with high Proteobacteria abundance had lower future-diabetes incidence ($p = 0.006$). At six weeks, isolated 116C-NOD mice with low Cyanobacteria YS2 genus and Tenericutes abundance exhibited reduced future-T1D incidence ($p = 0.002$ and $p = 0.010$, respectively). Moreover, isolated 116C-NOD with low Alphaproteobacteria RF32 genus abundance displayed reduced future-diabetes incidence at 12 weeks ($p = 0.012$) and 20 weeks ($p = 1 \times 10^{-4}$) compared to those with high abundance. Finally, across all three T1D-prone mice, those with low *Clostridium* (Ruminococcaceae family) abundance had significantly lower future-diabetes incidence compared to those with high abundance at 20 weeks ($p = 1 \times 10^{-4}$).

### 116C-NOD gut microbiota modulates the faecal bacterial community profile of NOD mice

The gut microbiota of isolated NOD, cohoused NOD, and isolated 116C-NOD mice were compared in a pairwise manner at six, 12, and 20 weeks of age when they were non-diabetic (independently of whether they eventually developed the disease), as well as at diabetes onset.

At six weeks of age, NOD cohoused with 116C-NOD mice exhibited a reduction of *Clostridium* (Lachnospiraceae family) (FDR $q = 0.015$), *Sutterella* ($q = 0.028$), Mogibacteriaceae unknown genus ($q = 0.053$),

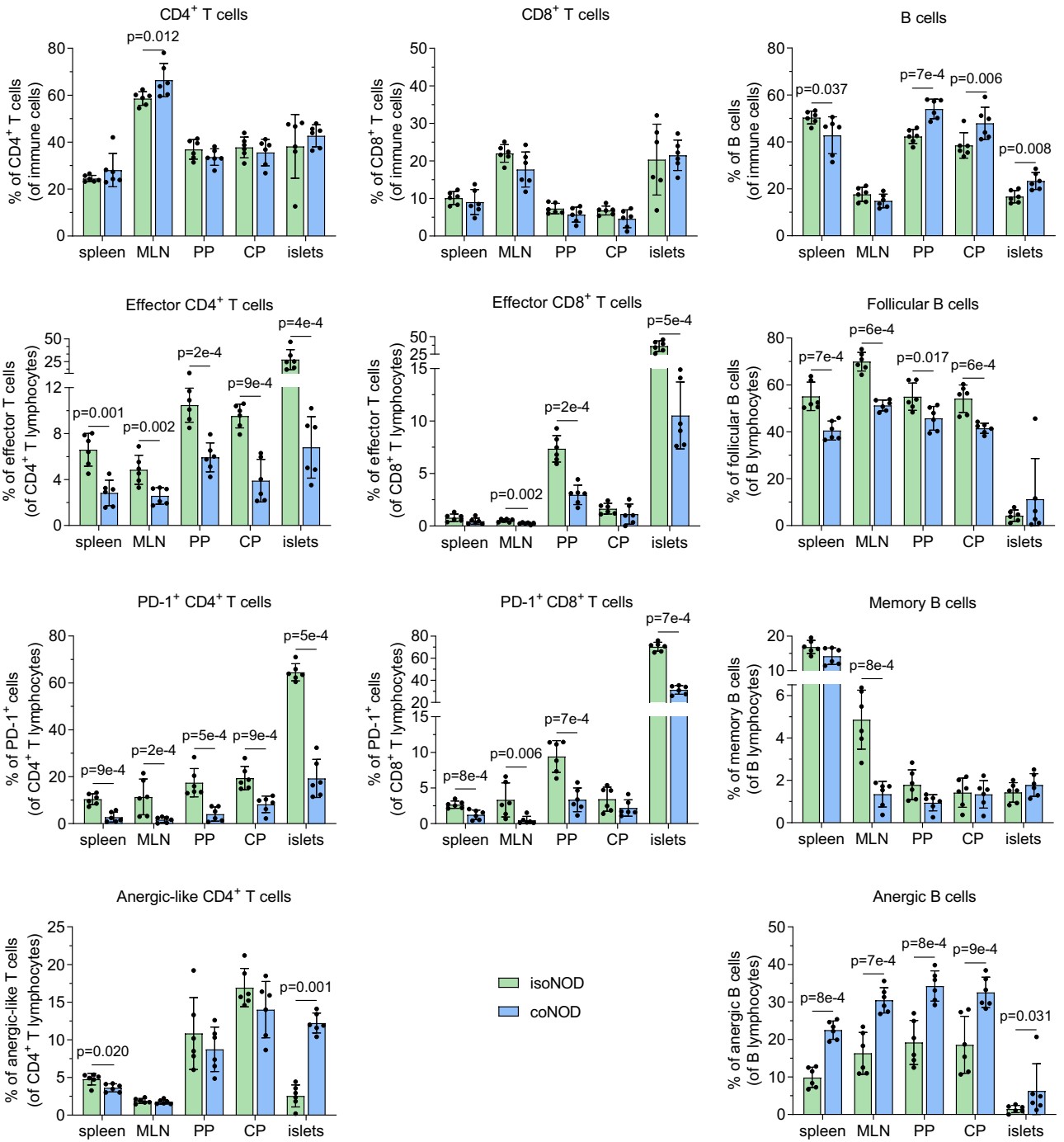

**Fig. 4 | T and B cell subsets of secondary lymphoid organs and pancreatic islet infiltrate from NOD mice isolated and cohoused with 116C-NOD mice.** Direct ex vivo immunophenotyping of lymphocyte subpopulations within spleen, mesenteric lymph nodes (MLN), Peyer's patches (PP), caecal patch (CP), and pancreatic islet infiltrate (islets), in NOD mice isolated (isoNOD) and cohoused (coNOD) ($n = 6$ for each organ and group of mice). The CD4[+] and CD8[+] T cell subsets included: effector T cells (CD44[high] CD62L[−] CD69[+] CD25[+]), exhausted-like T cells (PD-1(CD279)[+]) and anergic-like T cells (Foxp3[−] CD73[high] FR4[high]). The CD19[+] B220[+] B cell subsets comprised: follicular B cells (CD93[−] CD21[low] IgM[+] IgD[high] CD23[+]), memory B cells (CD38[high] CD138[−] GL-7[−]), and anergic B cells (CD93[−] CD21[low] IgM[−] IgD[high]). Two independent experiments were conducted (both represented). Data are expressed as mean ± SD and analysed with two-way ANOVA test (two-sided) on rank-transformed data-values.

and Cyanobacteria YS2 unknown genus ($q = 0.072$), compared with isolated NOD mice. These differences remained significant at 12 weeks only for *Sutterella* ($q = 0.044$). An increase in *Prevotella* ($q = 0.072$) in cohoused NOD mice at six weeks of age was also observed. Additionally, Actinobacteria were more represented in 116C-NOD mice at six weeks of age ($q = 0.052$). Contrary to expectations, in the comparison between cohoused NOD mice and isolated 116C-NOD, *Sutterella* and

Proteobacteria proportions were lower in the former at six weeks ($q = 0.002$ and $q = 0.036$, respectively) and 12 weeks of age ($q = 4 \times 10^{-4}$ and $q = 0.002$, respectively). Actinobacteria were also less abundant in cohoused NOD mice at six weeks ($q = 0.036$) (Fig. 7).

Furthermore, when involving the groups of future-resistant and future-diabetic mice in these analyses, we obtained the following results. At six and 12 weeks, future-diabetic mice of the cohoused NOD

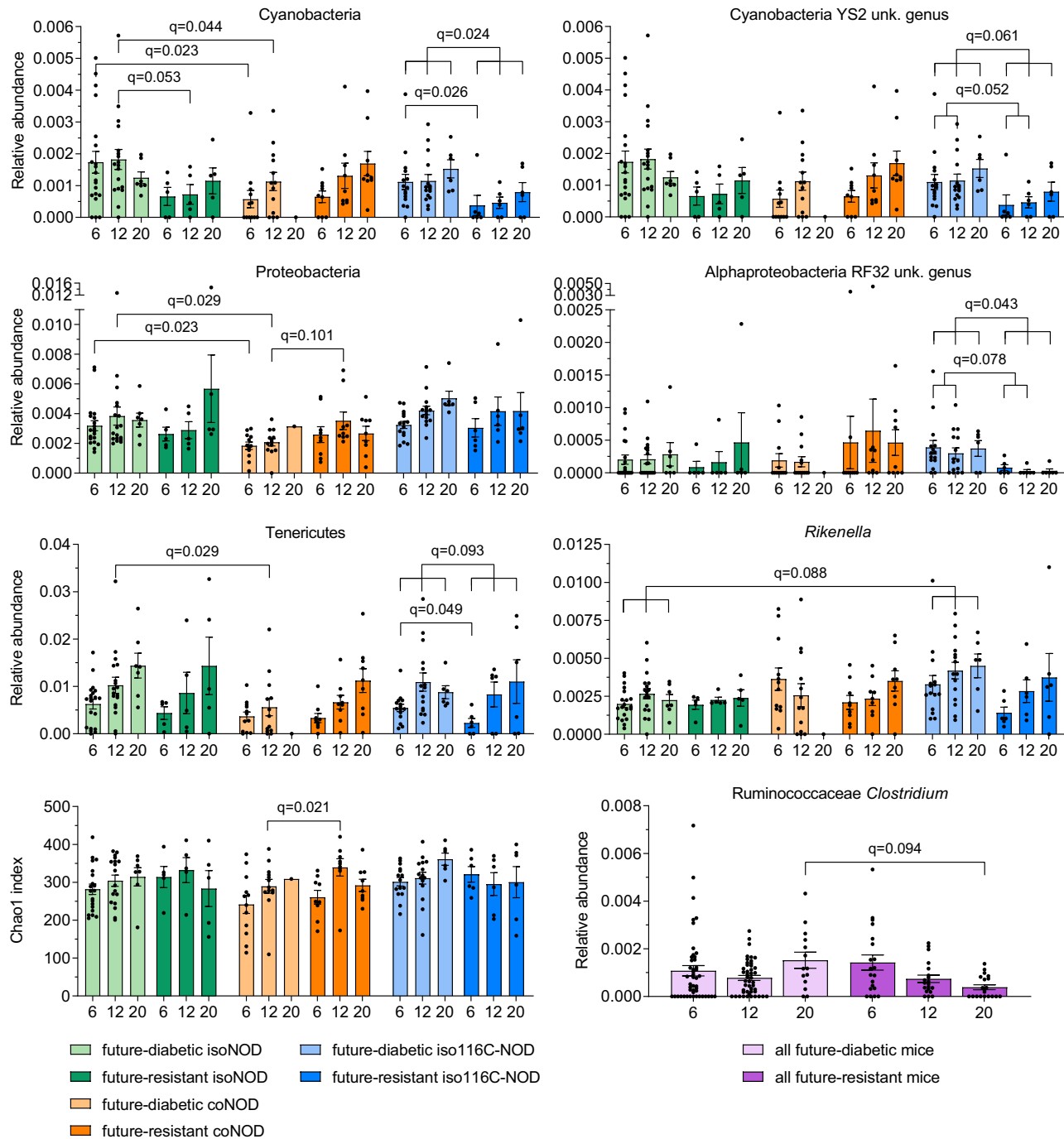

**Fig. 5 | Relative abundance of gut microbiota significant taxa and richness from future-diabetic and future-resistant T1D-prone mice.** 16S rRNA gene sequencing was conducted in faecal samples at 6, 12, and 20 weeks of age of isolated NOD (isoNOD), cohoused NOD (coNOD), and isolated 116C-NOD (iso116C-NOD). Animals were divided into two subgroups: future diabetics and future-resistant mice, which were compared within the same T1D-prone group and also amongst the three T1D-prone groups. Sample sizes: future-diabetic isoNOD (6 weeks: $n = 20$, 12 weeks: $n = 18$, 20 weeks $n = 7$); future-resistant isoNOD (6 weeks: $n = 5$, 12 weeks: $n = 5$, 20 weeks: $n = 5$); future-diabetic coNOD (6 weeks: $n = 12$, 12 weeks: $n = 13$, 20 weeks: $n = 1$); future-resistant coNOD (6 weeks: $n = 9$, 12 weeks: $n = 9$, 20 weeks: $n = 9$); future-diabetic iso116C-NOD (6 weeks: $n = 15$, 12 weeks: $n = 15$, 20 weeks: $n = 6$); future-resistant iso116C-NOD (6 weeks: $n = 6$, 12 weeks: $n = 6$, 20 weeks: $n = 6$). Data are expressed as mean ± SE. Relative abundance data were analysed using the MaAsLin2 statistical framework (mixed-effects linear regression model, two-sided test, adjustment for multiple comparisons), where $p$-values were corrected using the false discovery rate (FDR). Chao1 index was analysed with Mann–Whitney's test (two-tailed $p$-values).

group displayed lower levels of Proteobacteria ($q = 0.023$ and $q = 0.029$, respectively) compared with the future-diabetic group from isolated NOD. Tenericutes were also less abundant in cohoused NOD than in isolated NOD mice at 12 weeks ($q = 0.029$). Moreover, in isolated NOD mice, correlated samples of future-diabetic mice at six, 12, and 20 weeks of age displayed a lower relative abundance of *Rikenella*, compared to isolated 116C-NOD ($q = 0.088$) (Fig. 5).

Interestingly, the gut microbiota of diabetic animals from the isolated NOD group was enriched in Deferribacteres when compared to isolated 116C-NOD ($q = 0.054$). Moreover, diabetic mice from the

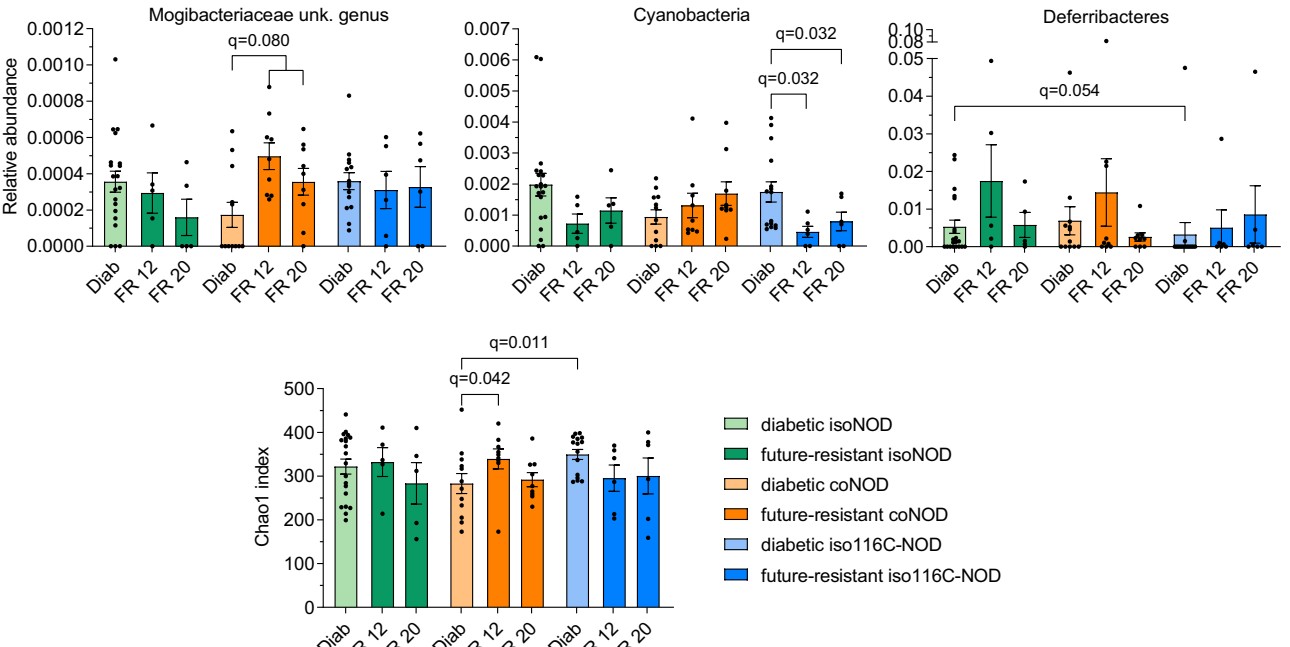

**Fig. 6 | Relative abundance and richness of gut microbiota significant taxa from diabetic and future-resistant mice.** 16S rRNA gene analysis was performed in faecal samples from diabetics (Diab) and future-resistant mice at 12 weeks (FR 12) and 20 weeks (FR 20) of isolated NOD (isoNOD), cohoused NOD (coNOD), and isolated 116C-NOD (iso116C-NOD) groups. Sample sizes: Diab isoNOD ($n = 20$); FR 12 isoNOD ($n = 5$); FR 20 isoNOD ($n = 5$); Diab coNOD ($n = 12$); FR 12 coNOD ($n = 9$); FR 20 coNOD ($n = 9$); Diab iso116C-NOD ($n = 15$); FR 12 iso116C-NOD ($n = 6$); FR 20 iso116C-NOD ($n = 6$). Data are expressed as mean ± SE. Relative abundance data were analysed using the MaAsLin2 statistical framework (mixed-effects linear regression model, two-sided test, adjustment for multiple comparisons), where $p$-values were corrected using the false discovery rate (FDR). Chao1 index was analysed with Mann-Whitney's test (two-tailed $p$-values).

isolated 116C-NOD group had a richer gut microbiota than cohoused NOD mice based on the Chao1 index ($p = 0.011$) (Fig. 6).

### T cell developmental disorders in young NOD and 116C-NOD mice enable the proliferation of segmented filamentous bacteria in their gut microbiota

Altogether, the results described above indicated the relevance of the intestinal microbiota in the modulation of the immune system as well as the effect of the immune system footprint on gut microbiota composition in murine autoimmune diabetes. To further explore this issue, faecal microbiota from NOD and 116C-NOD mice were compared to gut microbiota from immunodeficient NOD.RAG-2$^{-/-}$ and 116C-NOD.RAG-2$^{-/-}$ mouse models, as well as the non-T1D-prone C57BL/6J mice, at six, 12, and 20 weeks of age.

In NOD and 116C-NOD mice, segmented filamentous bacteria (SFB) (namely *Candidatus Arthromitus* in the sequencing database, but referring to the mammalian SFB *Candidatus Savagella*) were detected at six weeks of age and decreased over time until completely disappearing at 20 weeks of age (FDR $q = 1 \times 10^{-6}$ for isolated NOD and $q = 5 \times 10^{-7}$ for isolated 116C-NOD). In contrast, immunodeficient NOD.RAG-2$^{-/-}$ and 116C-NOD.RAG-2$^{-/-}$ mice maintained SFB throughout life ($q = 2 \times 10^{-10}$ for the comparison of isolated NOD versus NOD.RAG-2$^{-/-}$ at 20 weeks; $q = 4 \times 10^{-6}$ for the comparison of isolated 116C-NOD versus 116C-NOD.RAG-2$^{-/-}$ at 20 weeks). Interestingly, SFB was not detected at any age in C57BL/6J mice ($q = 2 \times 10^{-5}$ for the comparison of isolated NOD versus C57BL/6J at six weeks) (Fig. 8a).

To deepen the understanding of this phenomenon, the transfer of purified T or B cells separately, or total splenocytes, from NOD donors to NOD.RAG-2$^{-/-}$ recipients, was performed at six weeks of age. Only those mice that received purified T cells ($q = 3 \times 10^{-5}$ for the comparison with mice recipients of B cells; $q = 0.001$ for the comparison with NOD.RAG2$^{-/-}$ control spleen transfer), or total splenocytes ($q = 1 \times 10^{-4}$ for the comparison with B cells; $q = 0.004$ for the comparison with NOD.RAG2$^{-/-}$ control spleen transfer), evidenced an absence of SFB in

their gut microbiota, indicating that T cells prevented the proliferation of these bacteria (Fig. 8b).

Thus, collectively, these results indicated that T1D-prone models such as NOD and 116C-NOD displayed T cell disorders at early ages, which enhanced the proliferation of SFB. This scenario was exacerbated in the case of NOD.RAG-2$^{-/-}$ and 116C-NOD.RAG-2$^{-/-}$, which harboured this bacterial genus at all ages due to their lack of T cells.

### Colonisation of *Bifidobacterium* requires the presence of lymphocytes and is boosted in a non-diabetogenic *milieu*

The relative abundance of *Bifidobacterium* was clearly higher in C57BL/6J compared with isolated NOD mice at six (FDR $q = 2 \times 10^{-5}$), 12 weeks ($q = 3 \times 10^{-6}$), and 20 weeks of age ($q = 0.011$). On the opposite end, *Bifidobacterium* was not detected in NOD.RAG-2$^{-/-}$ ($q = 0.054$ for the comparison with isolated NOD mice at 12 weeks) nor 116C-NOD.RAG-2$^{-/-}$ mice ($q = 0.017$ for the comparison with isolated 116C-NOD mice at 12 weeks). Interestingly, among the T1D-prone animals, the 116C-NOD model had higher levels of *Bifidobacterium* compared with isolated and cohoused NOD mice ($q = 7 \times 10^{-4}$) (Fig. 8a). Therefore, the results suggested that the presence of lymphocytes was a requirement for *Bifidobacterium* colonisation in the murine gut, but the lymphocyte repertoire determined the abundance of *Bifidobacterium*. In this regard, an immunotolerant lymphocyte milieu, like that of C57BL/6J mice, provided the most appropriate environment for *Bifidobacterium* gut colonisation. To a lesser extent, a low-grade diabetogenic lymphocyte repertoire, like that of 116C-NOD mice, could also allow *Bifidobacterium* establishment.

The relative abundance of *Allobaculum* was also modulated by the diabetogenic lymphocyte context. They were found to be significantly less abundant in the gut microbiota of isolated NOD mice in comparison with C57BL/6J controls ($q = 9 \times 10^{-14}$ at six weeks, $q = 2 \times 10^{-9}$ at 12 weeks, and $q = 1 \times 10^{-5}$ at 20 weeks). However, the results suggested that, unlike *Bifidobacterium*, *Allobaculum* did not require T nor B lymphocytes, as it was present in NOD.RAG-2$^{-/-}$ and 116C-NOD.RAG-2$^{-/-}$

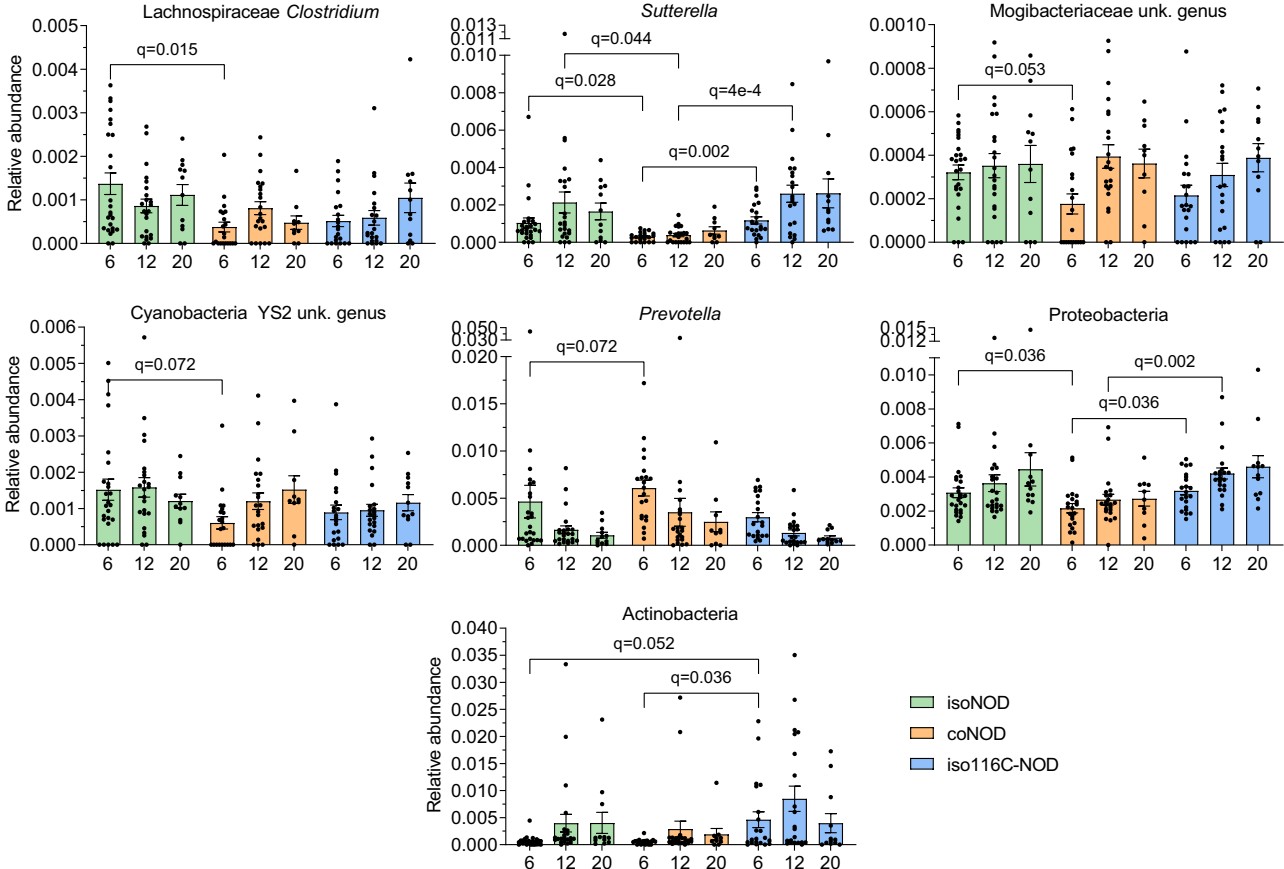

**Fig. 7 | Relative abundance of gut microbiota significant taxa from non-diabetic mice of the T1D-prone models.** 16S rRNA gene sequencing was conducted in faecal samples at 6, 12, and 20 weeks of age from non-diabetic isolated NOD (iso-NOD), cohoused NOD (coNOD), and isolated 116C-NOD (iso116C-NOD) mice. Sample sizes: isoNOD (6 weeks: $n = 25$, 12 weeks: $n = 23$, 20 weeks: $n = 12$); coNOD (6 weeks: $n = 21$, 12 weeks: $n = 22$, 20 weeks: $n = 10$); iso116C-NOD (6 weeks: $n = 21$, 12 weeks: $n = 21$, 20 weeks: $n = 12$). Data are expressed as mean ± SE. Relative abundance data were analysed using the MaAsLin2 statistical framework (mixed-effects linear regression model, two-sided test, adjustment for multiple comparisons), where $p$-values were corrected using the false discovery rate (FDR).

mice, and at even higher levels compared with isolated NOD and 116C-NOD mice at 20 weeks of age ($q = 0.087$ and $q = 0.042$, respectively) (Fig. 8a). These findings implied that the T1D-autoreactive lymphocyte repertoire inhibited the proliferation of *Allobaculum*.

**T and B cell deficiency determines a specific microbiome pattern in NOD.RAG-2$^{-/-}$ and 116C-NOD.RAG-2$^{-/-}$ mice**
Isolated NOD mice exhibited a higher microbial richness than NOD.RAG-2$^{-/-}$, as evidenced by the Chao1 index at 12 weeks ($p = 1 \times 10^{-4}$) and 20 weeks ($p = 0.027$) (Fig. 8c). Moreover, the intestinal microbiota profiles of isolated NOD and NOD.RAG-2$^{-/-}$ were significantly different at 12 weeks of age, as manifested by the β-diversity based on PERMANOVA test on unweighted (FDR $q = 0.001$) and weighted UniFrac distance ($q = 0.021$) indexes (Fig. 9a). In addition, the α-diversity of 116C-NOD gut microbiota was higher compared with 116C-NOD.RAG-2$^{-/-}$ mice at 12 weeks based on the Chao1 ($p < 1 \times 10^{-4}$) and Shannon ($p = 0.003$) indexes (Fig. 8c). The gut microbiota patterns of 116C-NOD and 116C-NOD.RAG-2$^{-/-}$ were significantly different at 12 and 20 weeks of age via the β-diversity PERMANOVA test on unweighted UniFrac distance index ($q = 0.001$) (Fig. 9b).

**116C-NOD B cells shape a diverse and enriched gut microbiota profile**
Regarding the relative abundances of bacterial taxa, 116C-NOD.RAG-2$^{-/-}$ were mainly enriched in *Adlercreutzia* and *Parabacteroides*. This was evidenced for *Adlercreutzia* at 20 weeks, when comparing 116C-

NOD.RAG-2$^{-/-}$ with NOD.RAG-2$^{-/-}$ mice (FDR $q = 0.002$). In addition, 116C-NOD.RAG-2$^{-/-}$ presented higher levels of *Adlercreutzia* than 116C-NOD mice at 12 weeks ($q = 0.013$) and 20 weeks of age ($q = 0.042$), as well as higher amounts of *Parabacteroides* than their immunocompetent counterparts at 12 weeks ($q = 0.049$). Moreover, NOD displayed a lower abundance of *Adlercreutzia* and *Parabacteroides* than C57BL/6 J mice at six weeks ($q = 0.017$) and 20 weeks of age ($q = 0.082$), respectively (Fig. 8d).

116C-NOD.RAG-2$^{-/-}$ mice exhibited a higher microbial richness than NOD.RAG-2$^{-/-}$ at 20 weeks, based on the Chao1 index ($p = 6 \times 10^{-4}$) (Fig. 8c). Also, the gut microbiota profiles of 116C-NOD.RAG-2$^{-/-}$ and NOD.RAG-2$^{-/-}$ were significantly different at 20 weeks of age, as shown by the β-diversity based on the PERMANOVA test on unweighted UniFrac distance index ($q = 0.002$), and at six weeks, based on weighted UniFrac distance index ($q = 0.070$) (Fig. 9c).

Overall, these findings indicated that 116C B cells defined a specific gut microbiota profile, favouring the colonisation of certain bacteria such as *Adlercreutzia* and *Parabacteroides*.

**The murine gut microbiota composition associated with genetic predisposition and resistance to autoimmune diabetes**
Isolated NOD mice exhibited a higher α-diversity than C57BL/6J, in particular, based on the Shannon index at six weeks ($p = 0.001$) and 12 weeks ($p < 1 \times 10^{-4}$), and the Chao1 index at 12 weeks of age ($p = 0.003$) (Fig. 8c). The gut microbiota patterns of isolated NOD and C57BL/6J mice were significantly different, as evidenced at all ages by

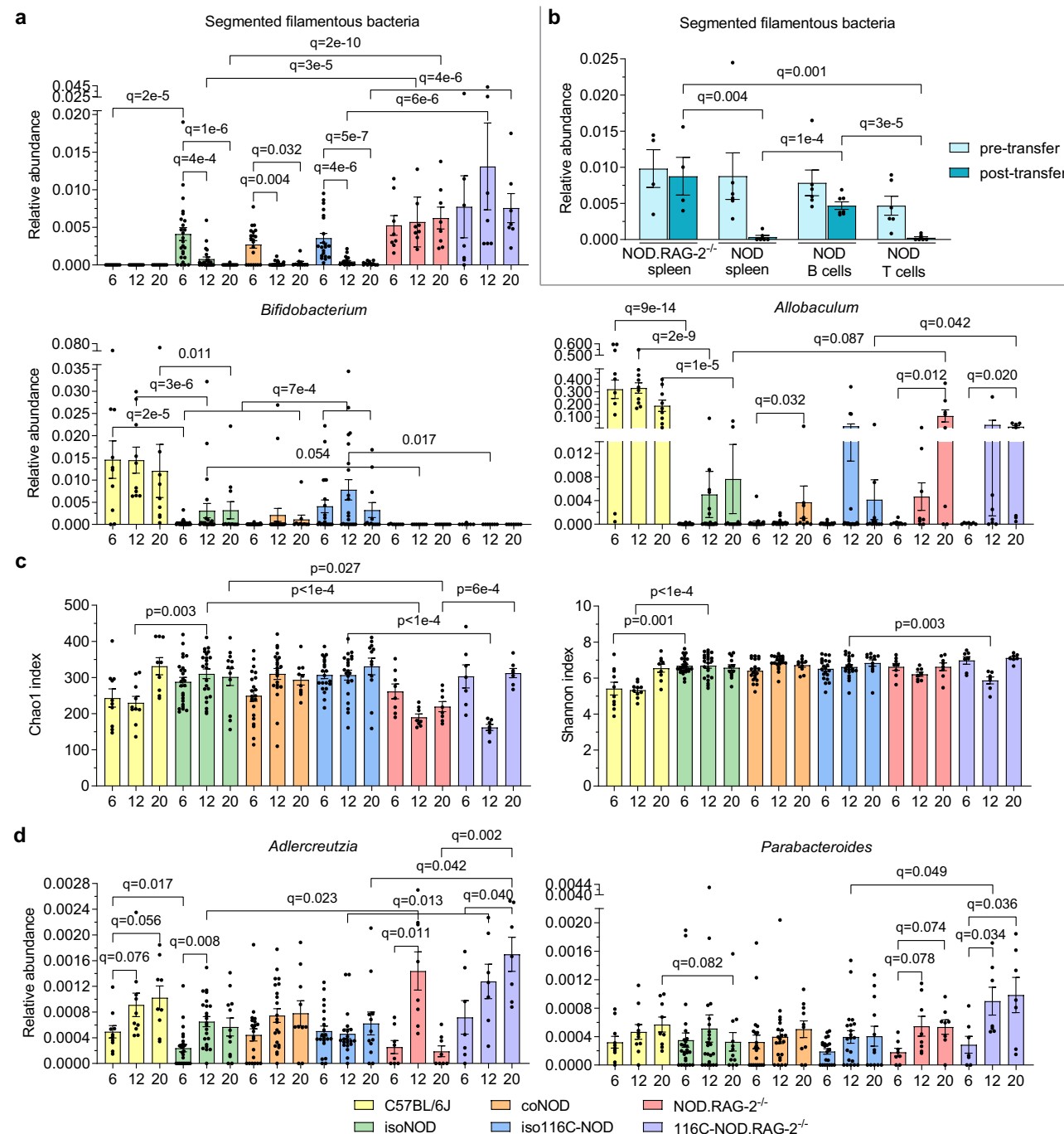

**Fig. 8 | Relative abundance and α-diversity of gut microbiota significant taxa from immunocompetent and immunodeficient mouse models.** 16S rRNA gene analysis was performed in faecal samples at 6, 12, and 20 weeks of age of different mouse strains: control C57BL/6J (6 weeks: *n* = 10, 12 weeks: *n* = 10, 20 weeks: *n* = 9), isolated NOD or isoNOD (6 weeks: *n* = 25, 12 weeks: *n* = 23, 20 weeks: *n* = 12), cohoused NOD or coNOD (6 weeks: *n* = 21, 12 weeks: *n* = 22, 20 weeks: *n* = 10), isolated 116C-NOD or iso116C-NOD (6 weeks: *n* = 21, 12 weeks: *n* = 21, 20 weeks: *n* = 12), NOD.RAG-2⁻/⁻ (6 weeks: *n* = 8, 12 weeks: *n* = 8, 20 weeks: *n* = 8), and 116C-NOD.RAG-2⁻/⁻ (6 weeks: *n* = 7, 12 weeks: *n* = 7, 20 weeks: *n* = 7). **a** Relative abundance of significant taxa altered by the autoimmune diabetes lymphocyte milieu.

**b** Relative abundance of segmented filamentous bacteria before (pre-transfer, 6 weeks) and after the transfer (post-transfer, 12 weeks) of total NOD spleen (*n* = 6), NOD B cells (*n* = 7), NOD T cells (*n* = 7), and control NOD.Rag2⁻/⁻ spleen (*n* = 4). **c** α-diversity based on Chao1 and Shannon indexes. **d** Relative abundance of bacteria affected by 116C-NOD B cells. **a**, **c**, and **d** share the same legend. Data are expressed as mean ± SE. Relative abundance data were analysed using the MaAsLin2 statistical framework (mixed-effects linear regression model, two-sided test, adjustment for multiple comparisons), where *p*-values were corrected using the false discovery rate (FDR). Chao1 and Shannon indexes were analysed with Mann–Whitney's test (two-tailed *p*-values).

the β-diversity based on PERMANOVA test on unweighted and weighted UniFrac distance indexes (FDR *q* = 0.001) (Fig. 9d).

*Prevotella* was more abundant in C57BL/6J mice at six (*q* = 0.047), 12 weeks (*q* = 0.002) and 20 weeks (*q* = 0.002). Furthermore, the

bacterial taxa *Turicibacter* and an unknown genus of the Desulfovibrionaceae family were detected almost exclusively in the gut microbiota of C57BL/6J mice (Supplementary Fig. 5a). For *Turicibacter*, this was shown by the comparison between isolated NOD and C57BL/6J

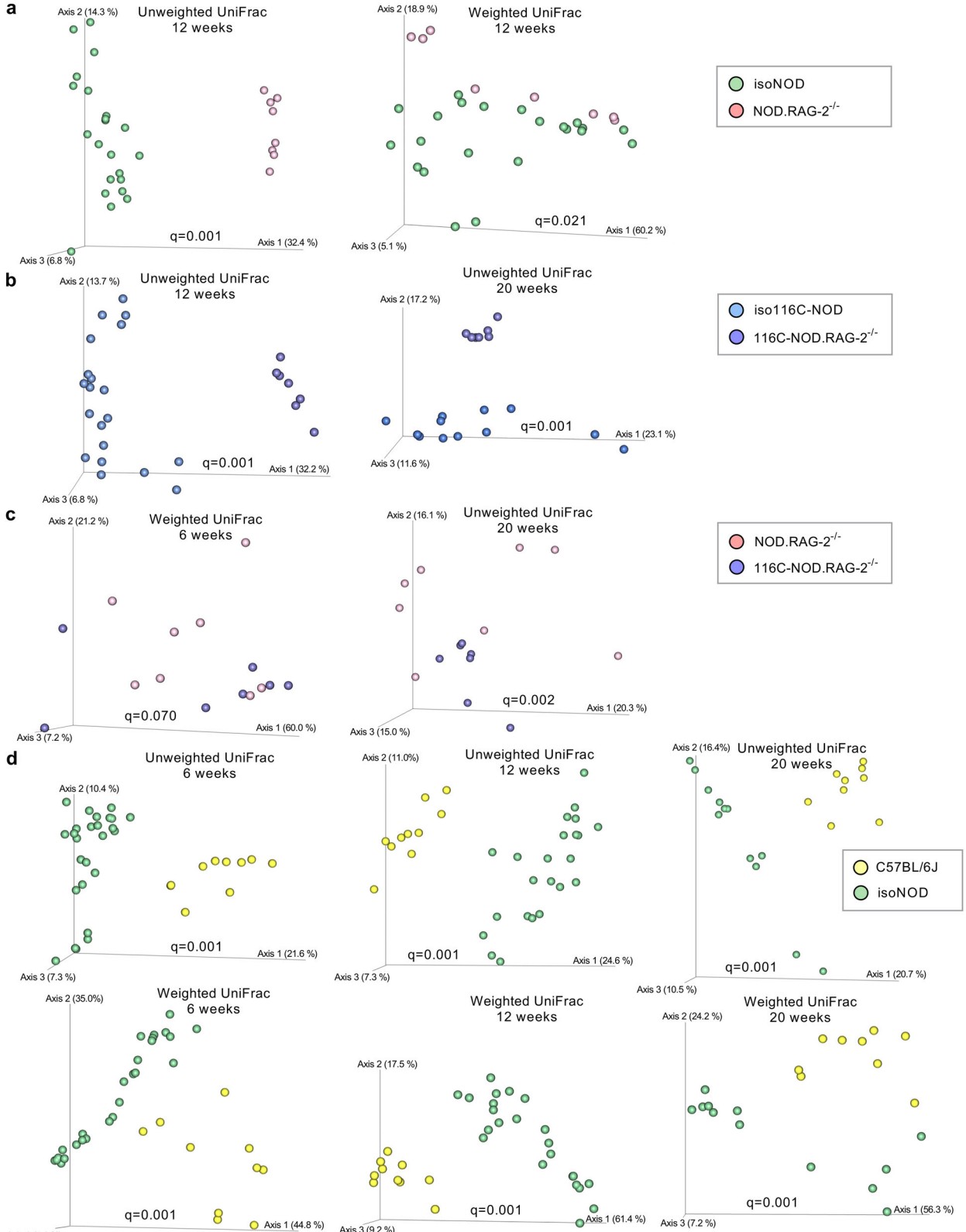

**Fig. 9 | β-diversity of gut microbiota from immunocompetent and immuno-deficient mouse models. a** Unweighted and weighted UniFrac distances at 12 weeks of isolated NOD (isoNOD) (n = 23) and NOD.RAG-2$^{-/-}$ (n = 8). **b** Unweighted UniFrac distances at 12 and 20 weeks of isolated 116C-NOD (iso116C-NOD) (n = 21 and n = 12, respectively) and 116C-NOD.RAG-2$^{-/-}$ (n = 7). **c** Weighted and unweighted UniFrac distances at 6 and 20 weeks of NOD.RAG-2$^{-/-}$ (n = 7) and 116C-NOD.RAG-2$^{-/-}$ (n = 8). **d** Unweighted and weighted UniFrac distances at 6, 12, and 20 weeks of C57BL/6 (n = 10) and isolated NOD (isoNOD) (n = 25, n = 23, and n = 12, respectively). Statistics were performed using the PERMANOVA test (two-sided), where p-values were corrected using the false discovery rate (FDR).

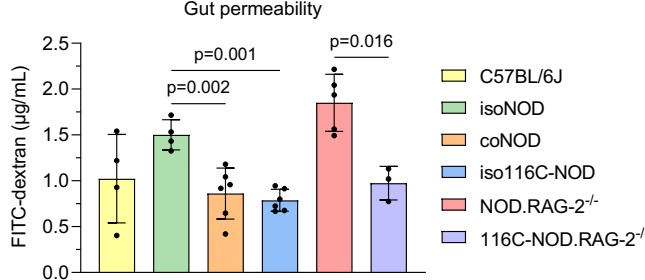

**Fig. 10 | Intestinal permeability of immunocompetent and immunodeficient mouse models.** In vivo FITC-dextran 4 kDa (FD4) gut permeability assay of the animal models at 12 weeks of age: control C57BL/6J ($n = 4$), isolated NOD or isoNOD ($n = 4$), cohoused NOD or coNOD ($n = 6$), isolated 116C-NOD or iso116C-NOD ($n = 6$), NOD.RAG-2$^{-/-}$ ($n = 5$), and 116C-NOD.RAG-2$^{-/-}$ ($n = 3$). FD4 was analysed in plasma samples. Two independent experiments were performed (both shown). Data are expressed as mean ± SD and analysed with two-way ANOVA test (two-sided) on rank-transformed data values.

mice at six weeks ($q = 1 \times 10^{-7}$), 12 weeks ($q = 1 \times 10^{-8}$), and 20 weeks ($q = 0.001$); and for the Desulfovibrionaceae genus, at 20 weeks ($q = 8 \times 10^{-5}$). Thus, these taxa were positively associated with genetic resistance to T1D. *Turicibacter* abundance augmented from six to 20 weeks in C57BL/6J mice ($q = 0.038$). Moreover, similar to SFB, *Prevotella* diminished from six to 20 weeks in immunocompetent mice ($q = 0.032$ for cohoused NOD and $q = 0.019$ for 116C-NOD mice). However, unlike SFB, this reduction over time was also observed in immunodeficient groups, suggesting that it was associated with the ageing.

The transfer of purified T cells, B cells, or total splenocytes from NOD donors to NOD.RAG-2$^{-/-}$ recipients at six weeks of age revealed the dependence of *Prevotella* on lymphocytes. Mice that received purified T cells harboured lower levels of this genus compared with mice that received B cells ($q = 0.050$) and NOD.RAG2$^{-/-}$ splenocytes ($q = 0.002$). Additionally, recipients of NOD spleen displayed a lower relative abundance of *Prevotella* than NOD.RAG2$^{-/-}$ controls ($q = 0.091$) (Supplementary Fig. 5b). Therefore, *Prevotella* colonisation was under T cell control.

Different bacterial taxa were negatively associated with genetic resistance to T1D or positively associated with a genetic predisposition to T1D (Supplementary Fig. 5c). This was exemplified by *Anaeroplasma*, which was more abundant in isolated NOD mice than in C57BL/6J mice at six weeks ($q = 0.002$), 12 weeks ($q = 5 \times 10^{-4}$), and 20 weeks of age ($q = 0.007$). Moreover, an unknown genus of the Rikenellaceae family was less abundant in C57BL/6J mice compared with isolated NOD at six weeks ($q = 9 \times 10^{-8}$), 12 weeks ($q = 1 \times 10^{-8}$), and 20 weeks of age ($q = 0.007$).

Supplementary Fig. 6 summarises the key findings of the study related to T1D incidence, immune profiling, and gut microbiota characterisation.

**Intestinal permeability is influenced both by gut microbiota and the immune system**

The assessment of gut permeability in all mouse groups was conducted to elucidate the underlying mechanism of the interaction between gut microbiota and the immune system in the context of autoimmune diabetes (Fig. 10).

Isolated NOD mice exhibited significantly higher gut permeability levels compared to cohoused NOD ($p = 0.002$) and isolated 116C-NOD mice ($p = 0.001$). Additionally, the intestinal permeability of 116C-NOD.RAG-2$^{-/-}$ mice was reduced compared to NOD.RAG-2$^{-/-}$ ($p = 0.016$). However, no differences were observed between 116C-NOD and 116C-NOD.RAG-2$^{-/-}$ mice.

Overall, the gut permeability assay indicated that the faecal microbiota communities of NOD mice in cohabitation with 116C-NOD and the gut microbiota of the 116C-NOD model could enhance the integrity of the intestinal barrier. Furthermore, this study strongly suggested that 116C-NOD-like B cells could play an important role in preserving the proper function of the gut barrier.

## Discussion

In humans and NOD mice, T1D develops following a mainly CD4$^+$ Th1 T cell-driven autoimmune response against pancreatic β-cells[4,5]. In a previous study, we demonstrated that B cells from 116C-NOD mice expressing a transgenic autoreactive BCR displayed an anergic phenotype. This phenotype led to a shift in the response of T lymphocytes towards Th17, which drove a decrease in the incidence of T1D[11]. Taking into account this background and the key role of faecal microbiota in the murine diabetic autoimmune process, the question that we posed next was whether the transfer of the gut microbiota of 116C-NOD mice could modulate the T1D incidence in NOD recipients. Consequently, the immunophenotype and the specific bacterial communities of gut microbiota from 116C-NOD, NOD mice cohoused with their transgenic siblings, and isolated NOD were analysed. Our findings highlighted several breakthroughs.

First, the transfer of faecal matter through cohousing or just in the presence of faeces from 116C-NOD mice in the cage was sufficient to decrease the T1D incidence in NOD mice. Cytokine secretion and expression of the major Th transcription factors mainly evidenced a shift from a Th1 towards an intermediate Th1/Th17 pattern in the cohoused NOD. This Th1/Th17 switch resembled the Th17 profile found in 116C-NOD mice[11]. Furthermore, cohoused NOD mice exhibited decreased levels of effector T cells in gut-associated lymphoid tissue (GALT) and pancreatic islet infiltrate, while showing increased levels of anergic-like CD4$^+$ T cells in pancreatic islets and anergic B cells in GALT. 16S rRNA gene sequencing revealed that 116C-NOD harboured a specific gut microbiota and, also, that the faecal microbiota of cohoused and isolated NOD mice significantly differed. Thus, taking these results together, we could assume that the immunophenotype of 116C-NOD mice shaped their gut microbiota pattern, and this 116C-NOD gut microbiota, when transferred to NOD mice, could drive changes in T and B cell responses and lymphocyte subpopulations within the GALT and pancreatic islet infiltrate, thereby decreasing the T1D incidence of the new host. This finding is in line with the work of Silverman et al.[20], in which the authors demonstrated that the protection from autoimmunity allowed by particular MHC alleles was possible through faecal microbiota transfer in NOD mice. Regarding the Th1/Th17 shift itself, this double-sided Th response was observed in prediabetic children with advanced β-cell autoimmunity[22].

Second, even though the diabetes incidence of cohoused NOD was similar to that of 116C-NOD and the microbiota community of cohoused NOD differed from that of isolated NOD, the colonisation of the donor microbiota (116C-NOD) was only partial, as the microbial communities of cohoused NOD and 116C-NOD presented differences for several bacterial groups. This observation is in agreement with our previous study showing that faecal transplantation of stools to rats was partial, although it improved clinical manifestations in the colitis rat model[23]. Therefore, the transfer of gut microbiota from 116C-NOD to NOD mice does not necessarily imply the implantation of the whole gut microbiota from the donor (which may not find a proper niche), but can induce a reshaping of other bacterial communities in the receptor. It is also conceivable that the initial composition of the microbiota and the presence of a diverse microbiota in weaned mice prior to the experiments could have influenced the extent of differences observed between the groups.

Third, we uncovered microbiomarkers of T1D in each of the three groups of diabetes-prone mice: isolated NOD, cohoused NOD, and 116C-NOD mice. Isolated NOD mice had Cyanobacteria as a marker,

being more abundant in future diabetics than in future-resistant animals. Cohoused NOD mice had a Proteobacterial marker, more abundant in future resistant-mice. The markers of 116C-NOD were Tenericutes and the unknown genera of Cyanobacteria order YS2 and Alphaproteobacteria order RF32, more abundant in future-diabetic mice. Therefore, the T1D bacterial fingerprint was dependent on the individual immune system of each T1D-prone mouse. However, a common T1D-bacterial marker was also found among the three groups; *Clostridium* (from the Ruminococcaceae family) was more abundant in future diabetics.

To further explore the effect of the adaptive immune system of NOD and 116C-NOD models on their gut microbiota, faecal samples from the immunodeficient variants NOD.RAG-2$^{-/-}$ and 116C-NOD.RAG-2$^{-/-}$, as well as from a non-T1D-prone mouse control, were also analysed. This study was complemented by assessing the gut microbiota of immunodeficient mice transferred with T and B lymphocytes. Our findings in this field highlighted several discoveries.

We found that segmented filamentous bacteria (SFB) were present in NOD and 116C-NOD mice at six weeks of age, but gradually declined and disappeared completely by 20 weeks. However, immunodeficient NOD.RAG-2$^{-/-}$ and 116C-NOD.RAG-2$^{-/-}$ mice continued to have SFB throughout their lifespan. In contrast, SFB was not detected in non-T1D-prone C57BL/6J mice at any age. We also demonstrated that T cells prevented SFB colonisation via a lymphocyte transfer experiment. Therefore, T cell developmental disorders were clearly a driver of SFB colonisation in young NOD and 116C-NOD as well as in immunodeficient NOD.RAG-2$^{-/-}$ and 116C-NOD.RAG-2$^{-/-}$ at all ages. In agreement with our results regarding SFB, these bacteria decreased throughout life in ICR (Institute of Cancer Research) mice. SFB were found in the ileum and caecum from two to seven weeks of age and then disappeared[24]. It is important to highlight that the NOD strain was generated from Cataract Shinogi (CTS) mice, an inbred substrain of ICR mice[3]. Thus, the sharing of a certain genetic background between NOD and ICR mice may indicate that both carry some immunological defects (possibly related to T cell development).

Even more interesting, the same study from Yin et al.[25] showed that SFB colonisation was also age-dependent in humans. SFB was present in human faecal samples until 36 months of age, with the highest occurrence (78.6%) observed between seven and 12 months. Fewer adults between 21 and 41 years (30.0%), and 41 and 51 years (16.7%), had SFB in their gut microbiota. Hence, this suggests that young NOD and 116C-NOD mice may resemble infants colonised with SFB, and these children may have lymphocyte development imbalances that are resolved as the immune system matures. Similarly, C57BL/6J mice could be equivalent to children without SFB, displaying a balanced lymphocyte development.

Furthermore, our findings indicated that the colonisation of *Bifidobacterium* required the presence of lymphocytes, and were particularly enhanced in a non-diabetogenic *milieu*. Our results align with numerous studies that have reported a decrease of *Bifidobacterium* within the gut microbiota of T1D patients[25,26–28].

The gut microbiota profiles of NOD and NOD.RAG-2$^{-/-}$ were significantly different. In addition, gut microbiota of NOD was richer than the faecal microbiota of NOD.RAG-2$^{-/-}$. These results are coherent with the lymphocyte status of both models. NOD mice have a rich T and B cell repertoire, while NOD.RAG-2$^{-/-}$ lack mature T and B cells. Hence, attending to the lymphocyte-dependence of certain bacterial communities to colonise the gut (as exemplified before by *Bifidobacterium*), the logical consequence of the absence of T and B cells is a less diverse gut microbiota in immunodeficient NOD.RAG-2$^{-/-}$. The same situation occurs between 116C-NOD and 116C-NOD.RAG-2$^{-/-}$, where both displayed significantly different intestinal microbiota patterns and 116C-NOD exhibited a more diverse microbiota than 116C-NOD.RAG-2$^{-/-}$. In this case, the difference was probably due to the

total absence of T cells and the presence of B cells with an anergic phenotype. In this sense, the anergic 116C B cells showed an enriched gut microbiota pattern in 116C-NOD.RAG-2$^{-/-}$ compared with NOD.RAG-2$^{-/-}$. Again, this fact is consistent with their lymphocyte status, since 116C-NOD.RAG-2$^{-/-}$ mice harbour B cells, which are absent in NOD.RAG-2$^{-/-}$. We found that 116C B cells could specifically induce the colonisation of bacteria such as *Adlercreutzia* and *Parabacteroides*. Interestingly, both genera were less abundant in the gut microbiota of multiple sclerosis patients[29]. Parallelly, in our work, NOD mice displayed lower levels of these genera compared with C57BL/6J controls. In contrast, in humans, *Parabacteroides* was associated with T1D onset[30].

Finally, we uncovered a connection between the gut microbiota and the immune system in their impact on intestinal permeability. Isolated NOD mice displayed higher levels of gut permeability compared to cohoused NOD and isolated 116C-NOD mice. Moreover, 116C-NOD.RAG-2$^{-/-}$ mice exhibited lower levels of gut permeability compared to NOD.RAG-2$^{-/-}$, but not to 116C-NOD mice. This suggests that intestinal permeability represents a potential underlying mechanism for the observed effects of the gut microbiota and the immune system. On the one hand, the gut microbiota communities of cohoused NOD mice may drive the immunological changes resulting in a reduction of T1D incidence through the enhancement of the intestinal barrier integrity. The preservation of the gut barrier prevents the translocation of bacteria and bacterial products that may have the potential to activate autoreactive lymphocytes in GALT. This would explain the higher proportions of effector T cells in the GALT of isolated NOD mice, which could migrate to pancreatic islets, where their levels are also elevated compared to cohoused NOD mice. In line with our results, Sorini et al.[31] observed that the disruption of the gut barrier can trigger the activation of islet-reactive T cells in the intestinal mucosa, leading to autoimmune diabetes in BDC2.5XNOD mice. Similarly, in humans, studies by Bosi et al.[32] and Sapone et al.[33] associated increased gut permeability to T1D. In addition, 116C-NOD B cells, along with those B cells with a similar phenotype, may contribute to the maintenance of a functional gut barrier, thereby influencing changes in the composition of the faecal microbiota. Therefore, gut permeability likely serves as a crucial mediator in the cross-talk between gut microbiota and the immune system in the context of autoimmune diabetes.

To gain a deeper understanding of the specific pathways and mechanisms connecting gut microbiota to T1D, the utilisation of advanced techniques of microbiota analysis such as shotgun DNA/RNA metagenomics or metabolomics approaches, along with direct ex vivo cytokine profiling analyses, are necessary.

Taken together, our results provide compelling evidence for the reciprocal interaction between the gut microbiota and the immune system in the T1D scenario. Importantly, this work could pave the foundation of a personalised microbiotherapy, tailored to an individual's immune status and gut microbiota profile, with the potential to serve as an innovative treatment strategy for T1D.

## Methods
### Ethical statement
Animal handling, maintenance and experimentation were conducted in accordance with the guidelines of the European Legislation for the Protection of Animals Used for Scientific Purposes. The animal procedures were approved by the Committee on the Ethics of Research in Animal Experimentation of the UdL (Protocol #: CEEA 06-02/19).

### Housing and husbandry
Mice were bred and maintained in the rodent animal house of the UdL. Animals were kept under specific pathogen-free (SPF) conditions and provided with autoclaved food (Envigo, Cat#2018S) and water *ad libitum*. The light-dark cycle was controlled in a 12:12 h format.

Temperature was set at 21 ± 2 °C, and relative humidity was held at 55 ± 5%. All the mice cages were located in the same SPF room. None of the mice received antibiotic treatment. Anaesthesia was induced and maintained via isoflurane inhalation at 4% and 2%, respectively. Mice were euthanized either by isoflurane inhalation or cervical dislocation when they developed diabetes or at the end of the studies.

## Mouse models
NOD mouse strain was originally purchased as NOD/ShiLtJ from The Jackson Laboratory (Bar Harbor, ME, Cat#JAX:001976). NOD.RAG-2[−/−] mouse strain was obtained from Dr. P. Santamaria (University of Calgary, Alberta, Canada)[8]. The 116C-NOD mouse model, transgenic for a β-cell-autoreactive B lymphocyte, was generated on the NOD strain genetic background in our laboratory[11]. The 116C-NOD.RAG-2[−/−] mouse model, which harbours monoclonal transgenic β-cell-autoreactive B cells, was generated on the NOD.RAG-2[−/−] strain genetic background by our team[11]. C57BL/6J mice were purchased from The Jackson Laboratory (Charles River, Europe, Cat#JAX:000664).

## Experimental design of animal groups
NOD and 116C-NOD mice were obtained from 116C-NOD breeder pairs (116C-NODxNOD). Weaned females were distributed in cages depending on two experimental conditions: isolation and cohousing between non-transgenic and transgenic siblings. Mice were grouped in cages of isolation which only contained NOD or 116C-NOD females, and in cages of cohousing with a balanced number of NOD and 116C-NOD littermates.

A cage change experiment was undertaken to confirm the results of the isolation and cohousing conditions. NOD females were housed in cages previously occupied by NOD or 116C-NOD female donor counterparts. The cages containing the donors' faecal pellets were renewed thrice weekly.

In parallel, NOD.RAG-2[−/−] and 116C-NOD.RAG-2[−/−] were obtained from 116C-NOD.RAG-2[−/−] breeder pairs (116C-NOD.RAG-2[−/−]xNOD.RAG-2[−/−]). In this case, weaned females were grouped in cages that only contained NOD.RAG-2[−/−] or 116C-NOD.RAG-2[−/−].

C57BL/6J female mice were used as non-T1D-prone controls.

Specific groups of mice were subjected to different in vivo and in vitro assays and were maintained until the appropriate age for those (see the sections below).

## Diabetes incidence assessment
Diabetes in female mice from NOD and 116C-NOD (isolation, cohousing, and cage change groups), NOD.RAG-2[−/−] and 116C-NOD.RAG-2[−/−] strains was followed up weekly and for ten months by measuring glycosuria with Medi-Test Glucose urine test strips (Macherey-Nagel, Cat#93001). Animals were considered diabetic after two consecutive positive readings of values ≥50 mg/dL and by monitoring glycemia levels with Accu-Chek Performa Glucose blood test strips (Roche, Cat#06454011) after obtaining values ≥250 mg/dL.

## Insulitis scoring
Pancreas from isolated and cohoused female NOD and 116C-NOD mice, aged six and 12 weeks, were embedded in Tissue Freezing Medium (Electron Microscopy Sciences, Cat#72592-C) and snap frozen in a ≤−75 °C cooling bath of dry ice and isopentane (Sigma-Aldrich, Cat#M32631). Pancreatic cryosections of 8 μm were rapidly collected on standard glass slides and immediately fixed in cold 95% ethanol for 10 min, dried for another 10 min, and frozen at −80 °C until hematoxylin and eosin (H/E) staining. H/E staining consisted of the following steps: thawing in air at 4 °C for 5 min and in PBS at 4 °C for another 5 min, staining for 5 min in filtered hematoxylin solution (5.3 g/L 1-hydrate Gurr (VWR Chemicals, Cat#340374T), 70.24 g/L Al$_2$(SO$_4$)$_3$ (VWR Chemicals, Cat#100103M), 300 mL/L glycerol, and 0.5 g/L sodium iodate (Honeywell, Cat# 71702)), washing in tap water for

5 min, differentiation in 0.5-1% HCl 70% ethanol solution and 0.001% ammonia solution, washing in tap water for another 5 min, staining in eosin solution (10 g/L eosin Y Gurr (VWR Chemicals, Cat#341972Q) and 80% ethanol) for 5 min, brief washing in tap water, dehydration in 80%, 85%, 90%, 95%, and 100% ethanol solutions (1 min in each), 3 min in absolute xylene, and finally mounting in DPX. Insulitis degree was determined by means of a blind analysis of 15–30 islets/mouse based on the following scoring criteria: score 0 (no cell infiltration in the islet), score 1 (peri-insulitis), score 2 (mononuclear cell infiltration in <25% of the islet), score 3 (mononuclear cell infiltration in 25-75% of the islet), and score 4 (mononuclear cell infiltration in >75% of the islet). The mean insulitis score of each pancreas was calculated via the equation: insulitis score = [(0 × A) + (1 × B) + (2 × C) + (3 × D) + (4 × E)]/ TNI]; where A, B, C, and D are the number of islets belonging to score 0, 1, 2, 3, and 4, respectively, and TNI is the total number of islets.

## Lymphocyte stimulation
Under sterile conditions, spleens harvested from 12-week-old isolated and cohoused female NOD mice (four animals per group) were mechanically disrupted with glass slide frosted ends in HBSS (Dutscher, Cat#X0509-500) containing 1% heat-inactivated foetal bovine serum or hiFBS (Gibco, Cat#10270106) and converted into single-cell suspensions by passing splenocytes through 40 μm nylon filters. T and B lymphocytes were then separately purified via negative selection using isolation kits specific for each population: Mouse Pan T Cell Isolation Kit II (Miltenyi Biotec, Cat#130-095-130) and Mouse B Cell Isolation Kit (Miltenyi Biotec, Cat#130-090-862); as well as the Auto-MACS Pro Separator magnetic cell sorter (Miltenyi Biotec, Cat#130-092-545), following manufacturer's instructions. Yield and purity of T and B cells were assessed by staining CD3 and CD19 cell surface markers with the monoclonal antibodies FITC anti-CD3 (BD Pharmingen, Cat#561798) at 2 μg/mL and violetFluor 450 anti-CD19 (Tonbo Biosciences, Cat#75-0193-U100) at 0.8 μg/mL in PBS with 1% hiFBS at 4 °C for 20 min, and by using the flow cytometer FACSCanto II (BD Biosciences). Lymphocyte purity was only accepted when values were greater than 90%.

Purified T and B lymphocytes were cultured in vitro with different stimuli at 37 °C and 5% CO$_2$ in complete culture media (CCM) consisting of RPMI 1640 (Biowest, Cat# L0501-500) supplemented with 10% hiFBS, 2 mM ʟ-glutamine (Corning, Cat#25-005-CI), 1 mM sodium pyruvate (Gibco, Cat#11360-070), 50 μM 2β-mercaptoethanol (Sigma-Aldrich, Cat#M6250-100ML), 100 U/mL benzylpenicillin sodium (Normon, Cat#602896.4), and 100 μg/mL streptomycin sulphate (Normon, Cat#624569.9).

T lymphocyte stimulation involved a three-day incubation of the purified T cells plated at 3 × 10$^5$ cells/well and cultured either: (1) alone; (2) with soluble anti-CD3 (sαCD3); (3) with well-coated anti-CD3 or fixed anti-CD3 (FαCD3); or (4) with sαCD3 in the presence of purified B cells, also plated at 3 × 10$^5$ cells/well (co-culture at a 1:1 ratio). Under the fourth condition, all possible combinations between T and B cells from isolated and cohoused NOD mice were made. When soluble, purified anti-CD3 monoclonal antibody (BD Pharmingen, Cat#553057) was added at 5 μg/mL. In the coated-plate condition, wells were incubated with 40 μL of 10 μg/mL anti-CD3 diluted in TBS buffer (pH = 9.4) at 37 °C for 2 h, left at 4 °C for ≤24 h and washed thrice with PBS before use. Important note: The two types of anti-CD3 stimulation, sαCD3 and FαCD3, exert a different effect on T cells. SαCD3 induces a low activation of T cells when these have no other stimuli in the culture. When sαCD3 is present in the co-culture of T and B cells, it acts as a linker between both lymphocytes, allowing their interaction. FαCD3 exerts a potent activation of T cells since the well-bound antibody molecules can cross-link T cell receptors (TCR).

Cell culture was performed in Nunclon Delta round-bottom 96-well plates (Nunc, Cat#163320) for all conditions, except for the coated-plate one, where Immulon 4 HBX flat-bottom 96-well plates

(Nunc, Cat#047612) were used. Cells and stimuli were added to a final CCM volume of 200 μl/well.

In vitro B lymphocyte stimulation was carried out by plating the purified B cells at $3 \times 10^5$ cells/well for two days in Nunclon Delta round-bottom 96-well plates and adding either: (1) no stimuli (ns), (2) lipo-polysaccharide (LPS) at 10 μg/mL (Sigma-Aldrich, Cat#L3012-5MG), (3) purified anti-B cell receptor or IgM (αBCR) monoclonal antibody (Jackson Immunoresearch, Cat#715-006-020) at 5 μg/mL, or (4) purified anti-CD40 (αCD40) monoclonal antibody (BD Pharmingen, Cat#553787) at 10 μg/mL plus IL-4 (R&D Systems, Cat#404-ML-010/CF) at 1 ng/mL, to a final CCM volume of 200 μl/well.

Stimulated lymphocytes were analysed in terms of cytokine profiling, transcription factor pattern, and proliferation index, as described below. Each study was performed from different plate wells as their fluorescent stainings were not compatible.

## Cytokine profiling

After lymphocyte purification and stimulation, culture supernatants were collected from both T and B cells assay plates and stored at −20 °C until use (for ≤1 week). The supernatants were screened for lymphocyte-derived secretion of IFN-γ, TNF-α, IL-17A, IL-6, IL-10, and IL-4 by means of the Cytometric Bead Array (CBA) Mouse Th1/Th2/Th17 Cytokine Kit (BD Pharmingen, Cat#560485), following manufacturer's instructions and acquiring samples in the flow cytometer FACSCanto II. Cytokine concentrations were calculated using the FCAP Array Software v3.0 (BD Biosciences).

## Proliferation assay

Following lymphocyte purification but prior to their stimulation, T and B cells were stained simultaneously with the cell division tracker carboxyfluorescein succinimidyl ester (CFSE) CellTrace (Invitrogen, Cat#C34554). The CFSE staining protocol was performed as follows. Cells were resuspended at $20 \times 10^6$ cells/mL in PBS at 37 °C. Protected from light, a stock solution of CFSE 50 μM was prepared just before use, also in PBS at 37 °C. The volume required to stain cells at 5 μM was gently added to the inner lateral wall of the sample microtube without touching the cell suspension. Immediately, the tube was closed, inverted and vortexed at low speed for 5 s to achieve optimal homogenisation of cells with CFSE. Lymphocytes were incubated for 2.5 min (never exceeding 3 min to avoid toxicity) in a 37 °C water bath with gentle agitation. Residual tracker was removed by incubating samples twice with five volumes of PBS supplemented with 10% hiFBS for 5 min at 37 °C. Cells were checked for CFSE staining by flow cytometry and subsequently stimulated in vitro following the previously described protocol. At the end of the lymphocyte cultures, CD4+ and CD8+ T cells were stained as mentioned before. Cell proliferation of T and B cells was analysed through CFSE-generational tracing in the FACSCanto II flow cytometer. Proliferation index was calculated using FCS Express 7.18.0015 (De Novo Software).

## Transcription factor analysis

The major transcription factors T-bet, GATA3, RORγT, and Foxp3 were analysed in T cell subpopulations after in vitro stimulation. CD4+ and CD8+ T cells were stained with the monoclonal antibodies PerCP anti-CD4 (BD Pharmingen, Cat#553052) at 0.8 μg/mL and PE anti-CD8 (BD Pharmingen, Cat#553033) at 0.8 μg/mL, in PBS with 1% hiFBS at 4 °C for 20 min. Intracellular staining of the transcription factors was achieved using the Foxp3/Transcription Factor Staining Buffer Set (eBioscience, Cat#00-5523-00), as well as the monoclonal antibodies PE-Cy7 anti-T-bet (eBioscience, Cat#25-5825-82) at 2 μg/mL, Alexa Fluor 488 anti-GATA3 (eBioscience, Cat#53-9966-42) at 0.5 μg/mL, APC anti-RORγT (eBioscience, Cat#17-6988-82) at 2 μg/mL, and eFluor 450 anti-Foxp3 (eBioscience, Cat#48-5773-82) at 2 μg/mL, following the instructions of the intracellular staining buffer set manufacturer. Samples were acquired with the flow cytometer FACS Canto II and the corresponding

FMO (fluorescence minus one) staining was used as a control. Flow cytometry data were analysed with FlowJo 10.0.7 (BD Biosciences).

## Direct ex vivo immunophenotyping of lymphocyte subsets

T and B cells from 12-week-old isolated and cohoused female NOD mice were phenotyped in spleen, mesenteric lymph nodes (MLN), Peyer's patches, caecal patch, and pancreatic islets (three animals per organ and group).

Firstly, the pancreatic islets were isolated by enzymatic digestion of the exocrine tissue as follows. The bile duct was exposed to clearly observe its anatomical connection with the liver and the duodenum. It was clamped near the hepatic duct using Schwartz micro-serrefines. Another clamp was applied at the Vater's ampulla using halsted-mosquito hemostats. The mouse was positioned under a binocular loupe and 3 mL of collagenase type IV (Worthington, Cat#LS004188) at a concentration of 0.8 mg/mL in HBSS (0.35 g/mL NaCOH₃) were slowly injected into the bile duct, near the clamped hepatic duct, using a 27 G x 10 mm butterfly needle. The distended pancreas was harvested and incubated for 30 min at 37 °C in a water bath to allow the digestion of the exocrine tissue. The enzymatic digestion was stopped by adding 10 mL of HBSS with 1% hiFBS at 4 °C. The pancreas was mechanically disrupted using a Pasteur pipette and transferred to a Petri dish to manually pick out the islets with a micropipette.

Secondly, Peyer's patches and the caecal patch were carefully excised from the gut using fine scissors under magnification. The spleen and MLN were harvested as described before without any additional procedures.

Finally, the isolated islets, spleen, MLN, Peyer's patches, and caecal patch were mechanically disrupted using glass slide frosted ends in HBSS with 1% hiFBS and then converted into single-cell suspensions by passing samples through 40 μm nylon filters.

The assessed CD4+ and CD8+ T cell subsets included: naïve T cells (CD44low CD62L+ CD69−), effector T cells (CD44high CD62L− CD69+ CD25+), effector memory T cells (CD44high CD62L− CD197−), central memory T cells (CD44high CD62L+ CD197+), tissue-resident memory T cells (CD44high CD62L- CD197- CD103+), exhausted-like T cells (PD-1(CD279)+ and LAG-3(CD223)+), and anergic-like T cells (Foxp3- CD73high FR4high). T cells were stained with the monoclonal antibodies: PerCP anti-CD4 (BD Pharmingen, Cat#553052) at 0.8 μg/mL, eFluor506 anti-CD8 (eBioscience, Cat#69-0081-82) at 2 μg/mL, APC anti-CD62L (BD Pharmingen, Cat#561919) at 2 μg/mL, BV421 anti-CD44 (Biolegend, Cat#103039) at 2 μg/mL, PE anti-CD69 (eBioscience, Cat#12-0691-81) at 1.6 μg/mL, BB515 anti-CD25 (BD Pharmingen, Cat#564458) at 1.6 μg/mL, PE-Cy7 anti-CD197 (Biolegend, Cat#120123) at 2 μg/mL, APC-Cy7 anti-CD103 (Biolegend, Cat#121431) at 2 μg/mL, BV421 anti-PD-1 (BD Pharmingen, Cat#562584) at 2 μg/mL, APC-Fire750 anti-LAG-3 (Biolegend, Cat#125240) at 2 μg/mL, efluor450 anti-Foxp3 (eBioscience, Cat#48-5773-82) at 2 μg/mL, PE-Cy7 anti-CD73 (eBioscience, Cat#25-0731-80) at 2 μg/mL, and APC anti-FR4 (BD Pharmingen, Cat#560318) at 2 μg/mL. Extracellular stainings were performed in PBS with 1% hiFBS at 4 °C for 30 min. Intracellular stainings were carried out following the protocol described in the "Transcription factor analysis" section.

The evaluated B cell subsets encompassed the following: follicular B cells (CD19+ B220+ CD93- CD21low IgM+ IgDhigh CD23+), marginal zone B cells (CD19+ B220+ CD93- CD21high IgMhigh IgDlow CD23−), T1 B cells (CD19+ B220+ CD93+ IgMhigh IgD+/− CD23−), T2 B cells (CD19+ B220+ CD93+ IgMhigh IgD+/− CD23+), anergic B cells (CD19+ B220+ CD93- CD21low IgM− IgDhigh), germinal centre B cells (CD19+ B220+ CD38low CD138- GL-7+), memory B cells (CD19+ B220+ CD38high CD138− GL-7−), plasmablasts (CD19+ B220+ CD138+), and plasmacytes (CD19- B220+ CD38low CD138+). B cells were stained with the monoclonal antibodies: BV510 anti-CD19 (Biolegend, Cat#115545) at 2 μg/mL, AlexaFluor 647 anti-B220 (Biolegend, Cat#103226) at 2 μg/mL, PE-Cy7 anti-CD93 (Biolegend, Cat#136505) at 2 μg/mL, PE anti-CD21 (Biolegend, Cat#123409) at 1.6 μg/mL, AlexaFluor 488 anti-IgM (Biolegend, Cat#406522) at 2 μg/

mL, PerCP anti-IgD (Biolegend, Cat#405736) at 2 µg/mL, BV421 anti-CD23 (BD Pharmingen, Cat#562929) at 2 µg/mL, APC-Fire750 anti-CD38 (Biolegend, Cat#102737) at 2 µg/mL, BV421 anti-CD138 (BD Pharmingen, Cat#566289) at 2 µg/mL, and PE-Cy7 anti-GL-7 (Biolegend, Cat#144619) at 2 µg/mL. Stainings were performed in PBS with 1% hiFBS at 4 °C for 30 min.

Samples were acquired with the flow cytometer FACS Canto II. Data were analysed with FlowJo 10.0.7.

## Lymphocyte transfer

Splenocytes, purified T cells, and B cells from 6-week-old NOD females were obtained following previously described methods (please see "Lymphocyte stimulation" section). Cells were washed twice with 0.9% NaCl and resuspended in the same solution. A volume of 100 µL containing $10 \times 10^6$ cells (either splenocytes, T lymphocytes or B lymphocytes) was intravenously injected into 6-week-old NOD.RAG-2$^{-/-}$ females through the retro-orbital sinus using 27 G × 13 mm needles. NOD.Rag2$^{-/-}$ donors were used as controls. The animals reached the end of the study at 12 weeks of age.

## 16S rRNA gene amplification and sequencing

Faecal samples from female NOD (isolated and cohoused), 116C-NOD, NOD.RAG-2$^{-/-}$, 116C-NOD.RAG-2$^{-/-}$ and C57BL/6 J mice were freshly collected from the same animals at six, 12, and 20 weeks of age as part of a longitudinal study. In the case of isolated NOD, cohoused NOD, and 116C-NOD mice, faecal samples at six, 12, and 20 weeks of age were collected from non-diabetic animals, including mice that later developed diabetes (future diabetics) and mice that remained resistant to the disease throughout the 40-week study period (future-resistant animals). Faecal samples were also isolated at any age of diabetic onset from isolated NOD, cohoused NOD, and 116C-NOD mice. Moreover, faecal samples of recipient NOD.RAG-2$^{-/-}$ females transferred with purified T cells, B cells, total NOD spleen, and total NOD.Rag2$^{-/-}$ spleen were also collected.

After collection, stool samples were stored at −80 °C until DNA extraction[23]. Genomic DNA was extracted following the recommendations of the International Human Microbiome Standards (IHMS; http://www.microbiome-standards.org)[34]. Briefly, a frozen aliquot (100 mg) of each sample was suspended in 250 µL of guanidine thiocyanate (Sigma-Aldrich, Cat# G6639), 40 µL of 10% N-lauryl sarcosine (Sigma-Aldrich, Cat# L9150), and 500 µL of 5% N-lauryl sarcosine. Mechanical disruption of the microbial cells with beads was applied, and nucleic acids were recovered from clear lysates by alcohol precipitation. The V4 hyper-variable region of the 16S rRNA gene was amplified by PCR for each sample using the following primers: forward (V4F_515_19: 5′-GTGCCAGCAMGCCGCGGTAA-3′) and reverse (V4R_806_20: 5′-GGACTACCAGGGTATCTAAT-3′) primers (Integrated DNA Technologies, custom primers)[35]. The amplicons were sequenced with the Illumina MiSeq system at the Autonomous University of Barcelona.

## Sequencing data analysis

The QIIME 2.0 bioinformatics pipeline was used to process the sequences. Briefly, sequences were demultiplexed, denoised, and dereplicated into amplicon sequence variants (ASVs) using the dada2 tool. Each sequence read was trimmed to a length of 298 bp. A total of 24,501,656 high-quality sequences of the 16S rRNA gene were generated from 360 faecal samples, with a mean of 68,060 sequences per sample. A feature table was generated for all samples with a minimum of 2710 sequences per sample. The feature table was then used to perform taxonomic classification and to measure α-and β-diversity. Taxonomy was assigned to each ASV using the 16S Greengenes database (gg_13_8_99 release), which contains 202,421 bacterial and archaeal sequences.

## Intestinal permeability assay

Gut permeability was assessed using FITC-dextran assay in 12-week-old female mice, including isolated NOD, cohoused NOD, isolated 116C-NOD, NOD.RAG-2$^{-/-}$, 116C-NOD.RAG-2$^{-/-}$, and C57BL/6J groups. At the start of the light cycle, animals were fasted for 4 h in cages with *ad libitum* access to water and no bedding. FITC-dextran 4 kDa (FD4) (TdB Labs, CAS#60842-46-8) was prepared freshly at a concentration of 80 mg/mL in PBS. It was administered via oral gavage at a dose of 7.5 µL/g of body weight using disposable PTFE 20 G × 38 mm feeding needles. A 5-min interval was allowed between the oral gavage of each mouse. After gavage, mice were transferred to new cages and kept under the previously mentioned conditions for an additional 4-h period. Blood samples were collected via cardiac puncture using 25 G × 25 mm needles, transferred to K3 EDTA microtubes (Sarstedt, Cat#41.1395.005), and kept at 4 °C while protected from light. The blood extraction was performed in the same order of mice and time interval between animals as the oral gavage. Plasma samples were obtained through two consecutive centrifugations at 4 °C (1500 × $g$, 10 min; 2500 × $g$, 15 min) and diluted 1/3 in PBS. A standard curve was prepared by serial dilutions of the remaining FD4 solution in PBS supplemented with 1/3 of plasma from PBS-gavaged mice. A volume of 100 µL of each diluted sample or standard was added to a black flat-bottom 96-well plate (Greiner Bio-One, Cat#655076). The concentration of FD4 in plasma samples was analysed using the fluorescence microplate reader Infinite M200 (Tecan) with an excitation wavelength of 493 nm and an absorption wavelength of 518 nm.

## Quantification and statistical analysis

Statistical parameters including the value of "n" (where n represents the number of mice per group), the expression of data with central values and dispersion measures (mean ± SE (standard error) or mean ± SD (standard deviation)), and the statistical methods applied to data are described in the corresponding figure legends.

GraphPad Prism 9.0.0 software was used to analyse survival curves of diabetes incidence with the Log-rank (Mantel-Cox) test (two-tailed and one-tailed $p$-values) and their hazard ratio with the Mantel–Haenszel test; as well as to compare groups of insulitis scores and the Chao1 and Shannon indexes with the Mann-Whitney test (one-tailed $p$-value for insulitis scores and two-tailed $p$-values for α-diversity indexes). Results with a $p$-value ≤ 0.05 were considered significant.

The cytokine profiling, transcription factor analysis, proliferation assay, direct ex vivo immunophenotyping of lymphocyte subsets, and intestinal permeability analysis were conducted in two independent experiments (the total number of samples, including both experiments, is shown in each graphic and figure legend). To assess reproducibility of the experiments, we performed two-way ANOVA tests on rank-transformed data-values to avoid violating normality and homoscedasticity assumptions of this parametric test; using mice/condition as one of the groups, and experiment (repetition 1 or repetition 2) as the second factor.

UniFrac distances were analysed through the PERMANOVA test. False discovery rate (FDR) corrected two-tailed $p$-values were taken into account to consider significant results ($q ≤ 0.10$). To determine the association between microbiome data and biological variables, we used linear mixed models as implemented in the MaAsLin2 (Microbiome Multivariable Association with Linear Models) package[36]. MaAsLin2 was set up with the following parameters: normalisation = "TSS", transform = "LOG", correction = "BH", analysis_method = "LM", max_significance = 0.25 (default significance threshold), min_abundance = 0.0001, min_prevalence = 0.1. Normalised taxa were modelled with a fixed effect of treatment group and random effects of time point and mouse ID. MaAsLin2 employs a mixed-effects linear regression model. The tests were two-sided and adjustments for multiple comparisons were performed. Results with a FDR $q ≤ 0.10$ were considered significant.

The entire list of materials and resources can be found as Supplementary Table 1.

**Reporting summary**

Further information on research design is available in the Nature Portfolio Reporting Summary linked to this article.

## Data availability

All sequence data generated in this study have been deposited in the NCBI database with the following access number: PRJNA989542. Source data are provided with this paper and are available in the Figshare data repository [https://doi.org/10.6084/m9.figshare.24459619.v1]. Source data are provided with this paper.

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

## Acknowledgements

This work was supported by the Plan Nacional de I + D + i of the Spanish Ministry of Science and Innovation (PID2019-109302RB-I00), the DiabetesCERO Foundation (Becas Impulso Talento Joven 2022), and CIBER of Diabetes and Associated Metabolic Diseases (CIBERDEM) that is an initiative from Instituto de Salud Carlos III (Spain). E.R.-M. was supported by predoctoral fellowships from the Generalitat de Catalunya (AGAUR FI-DGR, grant number: 2013FI_B 00585), the Spanish Government (FPU, grant number: FPU13/02045) and the IRBLleida. M.C.-P., B.A., and L.E.-M. were supported by UdL and IRBLleida predoctoral fellowships. F. Y. was supported by a predoctoral fellowship from the Chilean Government (ANID, grant number: 72190278). G.S.-G. was supported by a predoctoral fellowship from VHIR.

## Author contributions

Conceptualisation, E.R.-M., J.V. and C.M.; Methodology, E.R.-M., J.V. and C.M.; Formal Analysis, E.R.-M. and C.M.; Data curation, E.R.-M. and C.M.; Investigation, E.R.-M., A.S., M.C-P., F.Y., E.V., L.E.-M., B.A., C.C., A.P., G.S.-G and Co.M.; Writing – original draft, E.R.-M., J.V. and C.M.; Writing – review & editing, E.R.-M., J.V. and C.M.; Visualisation, E.R.-M.; Funding acquisition, J.V., C.M. and E.R.-M.; Supervision, J.V. and C.M.

## Competing interests

The authors declare no competing interests.
