## [Peer review file · Nature Communications]

REVIEWER COMMENTS

Reviewer #1 (expert in murine models of type-1 diabetes and gut microbiome):

This work investigates the modulation of the immune system by gut microbiota in type 1 diabetes models. Using NOD and NOD Rag2 ^{-/-} mice, and a previously established transgenic NOD mouse (116C-NOD) expressing β -cell-reactive Immunoglobulin on B-lymphocytes, the authors show effects of microbiota on T1D development and immune system, and reciprocally, the effect of immune cells on microbiota and T1D development.

Using two different co-housing approaches the authors find that microbiota affects diabetes incidence, either by alleviating or worsening T1D development. They show that microbiota manipulations (by co-housing or cage-change) modify T-cell cytokine and transcription factor profiles during in vitro stimulations, and lastly, present data on the role of T cells in regulating microbiota composition and, in inhibiting colonization of the gut with SFB (*C. arthromitus*).

The study supports a role for gut microbiota in the development of autoimmune diabetes in NOD mice and emphasizes its role in shaping T-cell immunity. While these themes have been investigated broadly over decades, the authors add among other things more detail to the interplay of T cells and SFB, and by using their own 116C-NOD mouse model, to the role of anergic B cells in microbiota composition.

Main comments:

In Figure 2, the role of different microbial taxa in diabetes development is given support by analyses in which mice are compared within the same study group (e.g. isolated mice or co-housed mice etc.) on the basis if they develop diabetes (future-diabetic) or not (future-nondiabetic mice). While there are significant differences between mice developing diabetes and mice not developing diabetes, the differences between co-housed or mixed and isolated mice are significant mostly between mice, which all develop diabetes (future-diabetic) irrespective of housing conditions. To evaluate if these differences are relevant for diabetogenesis, an age-related incidence graph should be drawn.

The relative proportions of bacterial genera shown in Figures 4 and 5 are mostly below 1 % and together cover less than 10% of the total microbiota. For example, the proportion of genus *Clostridium* (around 0.1 %) is surprisingly small as it is an abundant genus (between 10-30% in NOD mouse colonies in our experience). The composition of microbiotas should be described also with respect to the dominant genera, such as *Clostridium*, *Lachnoclostridium*, *Bacteroides* and *Prevotella*.

The co-housing was done by putting mice from two different groups to the same cage and the cage-change by putting mice to a cage previously inhabited by another group. In both instances, this involved mice already weaned, and the experiments therewith address microbiota changes, which may occur no earlier than at an age, in which the mice already possess a rich microbiota (i.e. after weaning). This may explain the small differences between microbiotas between study groups, and deserves to be mentioned (at least) in the Discussion.

Minor comments:

The complexity of the mouse groups grown under different conditions and the multitude of comparisons in the experiments impose on figure legends, which are very long, but this is probably necessary given their complexity.

In Results (line 291) it is stated that "NOD have a poorly-developed T cell status at early ages, which

allows the proliferation of SFB". This interpretation is not valid in light of the fact, that completely healthy Taconic mouse strains harbor SFB (including B6 mice).

The title could give a more succinct message than it now gives.

Reviewer #2 (expert in Immunology of type-1 diabetes):

Rossell-Mases et al., studying NOD mice, 11c-NOD mice and Rag2-deficient models of both show that the presence of T cells with their co-cultured B cell can influence Th-skewing and cytokine secretion. In addition, the authors show that the gut microbiota composition can be affected by age, diabetes development/resistance and the presence of T and B cells. The authors nicely show that immune cell transfers into Rag mice can modulate SFB abundance; however, there are a number of key experiments missing from this manuscript to confirm the findings and to move from association to causation, and the mechanism by which this occurs.

Main concerns:

Figure 2 – the authors assess the effect of T cell cytokines, but B cells are also present and they are measuring cytokines from supernatants – thus B cells may also be secreting these cytokines even though T cells are being directly stimulated. It would be good to show data from intracellular cytokine staining to confirm how much either T or B cells contribute to the secretion of cytokines – this could be coupled with transcription factor staining too. In line with this is T cell proliferation affected by the housing of iso or co B cells?

There is no directly ex vivo immunology data from the mice studied, only co-cultures, it would be good to study the T and B cell phenotyping in more detail in different tissues e.g. gut, pancreas, pancreatic lymph nodes to show in vivo evidence microbiota/housing directly effects the immune cells.

While the authors identify markers of prediction, the authors did not use these microbial markers to predict diabetes development – others have done this previously (<https://www.nature.com/articles/s41598-018-33571-z>) so this should be conducted to confirm their claim.

It is unclear how specifically the microbiota influence immune cells and how immune cells change microbiota. Investigation of mice treated with antibiotics, evaluating intestinal permeability etc may help provide more idea as to mechanism of action.

The lymphocyte transfers were conducted at a late age (6 weeks of age) when most Rag mice receiving transferred cells may not develop diabetes – did the authors observe diabetes incidences for the rag transfer studies and did this support the microbiota associations with diabetes development?

From Figure 4 onwards– the text seems to jump around from one figure to the next then back to the former, which got quite confusing when reading. My suggestion would be to move the figures around in the order presented in the text for easier reading.

Further methodological information should be provided e.g. Were mice studied housed in the same room? Were Rag2-/- mice treated with antibiotics? How many times were the experiments repeated and were the data reproducible?

Minor issues:

Islet infiltration – early not different but what about 40 weeks at the end of observation?

Figure 1 – make it easier to see the 0 incidence rates by separating the 2 lines on the x axis

Figure legend 1 – T is missing from T1D on line 668.

Line 158 – I am not sure TNFa produced by T cells was highly modulated by the presence of co-B cells as TNFa as the T cells have a bigger effect housed separately and only a small difference is seen between iso T cells +CoBs or isoTs – suggest removing this sentence

Line 188 – GATA3 does show a trend but I would remove mention of FoxP3 as that does not appear to.

Reviewer #3 (expert in gut microbiota and sequencing):

In this manuscript, Rosell-Mases et present the potential of the microbiota in the development of T1D as well as explore its interest as a predictive target.

The value of the work is of undoubted interest for public health as the authors conclude, as these findings open the door to personalized therapies in the next-future. While I do not doubt the high quality of the scientific work and hard work behind, I do believe that it would be fundamental that the authors look deep into the mechanisms by which the gut microbiota shape the T1D risk starting from these preclinical observations. In the current version, the analyses of the gut microbiota are restricted to 16S RRNA gene sequencing (metabolomics and metagenomics would be ideal) and there are no analyses that tackle the host-microbiota cross-talk. For instance, does the gut microbiota of NOD cohoused with 116C-NOD influence on the intestinal immune with consequences in the autoimmune response? If so, which bacterial metabolites rule are mediators? Or maybe any structural compound from bifidobacterial strains?

Mayor comments:

1. Authors should justify the decision of considering FDR lower than 0,10 as significant.
2. Authors refer to a-diversity, based on Chao1, which is more "richness". It is true that sometimes authors refer to Shannon (diversity index), but not always, like in lines 440-441, where they claim lower or higher diversity, when only the Chao1 index changes, but the Shannon index is the same.
3. Why the gut microbiota of cohoused 116C-NOD was not analyzed (Fig4)? How author explain the absence of differences between NOD and 116C-NOD (Fig 6)?
4. Authos described that CoNOD has low levels of Sutterella and Proteobacteria acquire from cohousing as a protective microbe against T1D (393-401). However, these genera have the same prevalence in isoNOD and iso116C-NOD. How do authors justify this claim?
5. Regarding the use of the microbiota as a possible predictive marker. Authors compare at 12 weeks, when they are already starting to develop diabetes, and not at 6 weeks. However, I think it would be interesting to compare also at 6 weeks as this time point might pinpoint better mice more "prone" to develop diabetes.
5. Do they confirm the differences found in vitro in vivo in the cytokine levels?
- 6- Can the authors provide any regarding th soundness of their hypothesis in human data?

Point-by-point responses to the reviewers:

Dear Reviewers,

We wish to thank the referees for the time spent reviewing our manuscript and appreciate the opportunity to address your comments and criticisms. In this regard, we feel that all the points they raised have allowed us to improve the overall quality and readability of the manuscript. Our answers are written in blue below the comments made by the referees. The modifications made to the manuscript have been added with track changes and colour highlighting. We hope that you now find the new version suitable for publication.

Reviewer #1 (expert in murine models of type-1 diabetes and gut microbiome):

This work investigates the modulation of the immune system by gut microbiota in type 1 diabetes models. Using NOD and NOD Rag2 ^{-/-} mice, and a previously established transgenic NOD mouse (116C-NOD) expressing β -cell-reactive Immunoglobulin on B-lymphocytes, the authors show effects of microbiota on T1D development and immune system, and reciprocally, the effect of immune cells on microbiota and T1D development.

Using two different co-housing approaches the authors find that microbiota affects diabetes incidence, either by alleviating or worsening T1D development. They show that microbiota manipulations (by co-housing or cage-change) modify T-cell cytokine and transcription factor profiles during in vitro stimulations, and lastly, present data on the role of T cells in regulating microbiota composition and, in inhibiting colonization of the gut with SFB (*C. arthromitus*).

The study supports a role for gut microbiota in the development of autoimmune diabetes in NOD mice and emphasizes its role in shaping T-cell immunity. While these themes have been investigated broadly over decades, the authors add among other things more detail to the interplay of T cells and SFB, and by using their own 116C-NOD mouse model, to the role of anergic B cells in microbiota composition.

Main comments:

In Figure 2, the role of different microbial taxa in diabetes development is given support by analyses in which mice are compared within the same study group (e.g. isolated mice or co-housed mice etc.) on the basis if they develop diabetes (future-diabetic) or not (future-nondiabetic mice). While there are significant differences between mice developing diabetes and mice not developing diabetes, the differences between co-housed or mixed and isolated mice are significant mostly between mice, which all develop diabetes (future-diabetic) irrespective of housing conditions. To evaluate if these differences are relevant for diabetogenesis, an age-related incidence graph should be drawn.

First of all, we understand that the reviewer referred to Fig. 4 (current Fig. 5) instead of Fig. 2, since the latter does not contain any data of microbial taxa from future-diabetic and future-resistant mice. We appreciate the reviewer's suggestion and as a result, we provide an additional age-related incidence analysis (please see Supplementary Fig. 2).

In this new analysis, we selected the significant bacterial taxa and Chao1 index for the diabetes prediction. Then, the three groups of future-diabetic and future-resistant T1D-prone mice (isolated NOD, cohoused NOD and isolated 116C-NOD) were classified into two subgroups: mice with high relative abundance of the corresponding taxa or high richness, and mice with low relative abundance of the corresponding taxa or low richness. The T1D incidence of these subgroups was compared. For detailed information, refer to the corresponding section of Results (lines 259-280).

Supplementary Fig. 2 | T1D incidence of NOD and 116C-NOD mice classified by their relative abundance of gut bacterial taxa and richness related to diabetes prediction.

Future-diabetic and future-resistant isolated NOD (isoNOD), cohoused NOD (coNOD) and isolated 116C-NOD (iso116C-NOD) were divided into two subgroups: mice with high relative abundance (RA) of the corresponding bacterial taxa or high richness, and mice with low relative abundance of the corresponding bacterial taxa or low richness. Diabetes incidence curves were analysed with the Log-rank (Mantel-Cox) and one-tailed t-test, where * $p \leq 0.05$, ** $p \leq 0.01$, *** $p \leq 0.001$, and **** $p \leq 0.0001$.

The relative proportions of bacterial genera shown in Figures 4 and 5 are mostly below 1 % and together cover less than 10% of the total microbiota. For example, the proportion of the genus *Clostridium* (around 0.1 %) is surprisingly small as it is an abundant genus (between 10-30% in NOD mouse colonies in our experience). The composition of microbiotas should also be described with respect to the dominant genera, such as *Clostridium*, *Lachnoclostridium*, *Bacteroides* and *Prevotella*.

Indeed, the proportion of *Clostridium* appears to be low in this particular analysis. *Clostridium* is a multiphyletic genus found in the most common 16S databases. It is annotated as belonging to several families, including Lachnospiraceae, Ruminococcaceae, and Clostridiaceae. The genus referred to by the reviewer is from the Lachnospiraceae family. This issue is related to a taxonomic annotation problem intrinsic to these public databases. Despite this annotation issue, we still can conclude that one of the species from this genus is different between the groups analyzed, but we cannot make this claim as species level analysis is not the most recommended based on 16S rRNA data. Actually, *Clostridium* has been recognized as the most problematic taxon in terms of classification (<https://www.microbiologyresearch.org/content/journal/ijsem/10.1099/ijsem.0.003698>). To overcome this issue, we have recently started to build a new 16S database that is correcting these heterogenous annotations using the NCBI taxonomy as reference; this work is not yet published, but will definitely be useful to other scientists in this microbiome field. This new database will still need further improvement and validations and cannot be used in this study yet.

Related to the second comment of the reviewer, we used the MaAslin2 R package, which returns the bacterial genera significantly different between the groups analyzed, to perform differential abundance analysis. From these analyses, the most dominant genera were not found differentially abundant in group comparisons. Also, given the high number of genera (more than 40 genera) detected per sample, we decided to plot only those found significantly different. This approach allows for a clearer visualization of the key findings and avoids overwhelming the plot with a large number of genera that do not show significant differences. This is consistent with the standard practices followed in the analysis of microbial abundance and visualizing data.

The co-housing was done by putting mice from two different groups to the same cage and the cage-change by putting mice to a cage previously inhabited by another group. In both instances, this involved mice already weaned, and the experiments there address microbiota changes, which may occur no earlier than at an age in which the mice already possess a rich microbiota (i.e. after weaning). This may explain the small differences between microbiotas between study groups, and deserves be mentioned (at least) in the Discussion.

In acknowledgment of the mice being already weaned and having diverse microbiota prior to the commencement of the study, we have included the following statement in the discussion section.

Discussion:

“...It is conceivable that the initial composition of the microbiota and the presence of a diverse microbiota in weaned mice prior to the experiments could have influenced the extent of differences observed between the groups...”

Minor comments:

The complexity of the mouse groups grown under different conditions and the multitude of comparisons in the experiments impose on figure legends, which are very long, but this is probably necessary given their complexity.

Indeed, as the reviewer points out, certain figure legends may be lengthy due to the complexity of the studies. However, we ensure that the word count of our figure legends remains within the prescribed limit set by the journal (<350 words). The longest legend has 283 words.

In Results (line 291) it is stated that “NOD have a poorly-developed T cell status at early ages, which allows the proliferation of SFB”. This interpretation is not valid in light of the fact, that completely healthy Taconic mouse strains harbor SFB (including B6 mice).

We appreciate the reviewer's comment and concur with the necessity to amend this sentence, considering the fact that C57BL/6 mice from vendors other than the Jackson Laboratory may harbour SFB. Previous studies have shown that Jackson Laboratory C57BL/6 mice exhibit lower levels of SFB or even an absence of these bacteria compared to Taconic C57BL/6 (Ivanov *et al.* 2008, <https://pubmed.ncbi.nlm.nih.gov/18854238/>).

The C57BL/6J and NOD mice used in our studies (with the latter being the strain in which the 116C-NOD, NOD.RAG-2^{-/-}, and 116C-NOD.RAG2^{-/-} models were generated) were purchased from The Jackson Laboratory and possessed the same specific pathogen-free (SPF) health status.

We rely on The Jackson Laboratory as our trusted mouse supplier due to their ability to maintain a stable gut microbiome: “...**Gut Microbiomes of JAX C57BL/6 Substrains are More Similar to Each Other than Other Vendors Across Locations and Health Status.** As has been published previously, mice from different vendors exhibited distinct microbial profiles at intake. However, JAX mice were observed to display a more similar microbiome when compared to mice from other vendors. Indeed, when visualized by Principle Coordinate Analysis, JAX mice cluster tightly together regardless of facility location or health status. Interestingly, the consistency of JAX microbiomes also held true when comparing C57BL/6NJ to C57BL/6J mice, two distinct C57BL/6 substrains, which have been genetically separated since 1951...” (<https://www.jax.org/news-and-insights/jax-blog/2019/August/microbiome-stability>).

While Taconic Biosciences' health standards permit the presence of SFB in certain cases, their guide states the following: “...While SFB is only one component of the gut microbiome, for many models the impact could be significant. For that reason, Taconic includes SFB on its health reports and **SFB is specifically excluded in its EF (excluded flora) health status...**” (<https://larc.ucsf.edu/sites/larc.ucsf.edu/files/wysiwyg/Taconic-Biosciences-2019-Pricing-Guide.pdf>)

Our results evidenced an increase in the relative abundance of SFB in young NOD, 116C-NOD, and, consistently at all ages, in immunodeficient NOD.RAG-2^{-/-} and 116C-NOD.RAG-2^{-/-} mice. Therefore, based on our findings and the aforementioned information, we have made the following correction to the sentence: “...**T1D-prone models such as NOD and 116C-NOD display T cell disorders at early ages, which enhance the proliferation of SFB.**” Moreover, we have removed the consecutive sentence: ~~Conversely, the well-developed T cell pool of C57BL/6J mice completely inhibits the colonisation of SFB from their youth (lines 335-340).~~

Thus, with these revisions, we emphasize that T cell disorders can facilitate SFB colonization, while not disregarding the potential for SFB proliferation in a well-developed T cell state.

The title could give a more succinct message than it now gives.

We can propose the titles below; however, we believe that the initial title was already succinct enough.

“Gut Microbiota-Immune System Modulation in Type 1 Diabetes Models”

“Modulation of Gut Microbiota-Immune System Interplay in Type 1 Diabetes Models”

Reviewer #2 (expert in Immunology of type-1 diabetes):

Rossell-Mases et al., studying NOD mice, 116c-NOD mice and Rag2-deficient models of both show that the presence of T cells with their co-cultured B cell can influence Th-skewing and cytokine secretion. In addition, the authors show that the gut microbiota composition can be affected by age, diabetes development/resistance and the presence of T and B cells. The authors nicely show that immune cell transfers into Rag mice can modulate SFB abundance; however, there are a number of key experiments missing from this manuscript to confirm the findings and to move from association to causation, and the mechanism by which this occurs.

Main concerns:

Figure 2 – the authors assess the effect of T cell cytokines, but B cells are also present and they are measuring cytokines from supernatants – thus B cells may also be secreting these cytokines even though T cells are being directly stimulated.

It would be good to show data from intracellular cytokine staining to confirm how much either T or B cells contribute to the secretion of cytokines – this could be coupled with transcription factor staining too. In line with this is T cell proliferation affected by the housing of iso or co B cells?

We acknowledge the reviewer’s concern regarding the secretion of cytokines and provide the following explanation to address and clarify this matter:

The secretion of cytokines IL-17A, IFN-gamma and IL-4 is not observed in B cells as indicated in Supplementary Fig. 3. *In vitro* cultures of B cells do not demonstrate the presence of IL-17A and IFN-gamma. Furthermore, B cells do not produce IL-4, and it is consumed equally by isoNOD B cells and coNOD B cells in the anti-CD40 plus IL-4 condition (as shown by the IL-4 control). Therefore, these three cytokines must be produced by T cells in the co-culture (Fig. 2a).

In the B cell *in vitro* culture (Fig. 2b), we observe the secretion of IL-6 and IL-10 by B lymphocytes upon stimulation with LPS. However, when stimulated with anti-BCR nor anti-CD40 plus IL-4 (very low levels of IL-10 in anti-CD40 plus IL-4 condition). Consequently, it can be inferred that T cells are responsible for the production of IL-6 and IL-10 in the co-culture, as illustrated in Fig. 2a.

Regarding the cytokine TNF-alpha, since it is secreted by B cells when these are stimulated with anti-BCR; in the co-culture, this cytokine could be produced by both T cells and B cells. Consequently, we have removed TNF-alpha from Fig. 2a. However, TNF-alpha is not an essential cytokine for our experiments. Its removal from the results does not affect the final interpretation.

The primary cytokines of focus in this study are IL-17A, IL-6 and IFN-gamma, which are exclusively produced by T cells in the co-culture with B cells (Fig. 2a).

Based on the aforementioned reasons, we believe that conducting additional experiments to distinguish the source of cytokines is unnecessary.

Supplementary Fig. 3 | *In vitro* secretion of IL-17A, IFN- γ and IL-4 by B lymphocytes from NOD mice isolated and cohoused with 116C-NOD mice. B cells from NOD mice isolated (isoNOD Bs) and cohoused (coNOD Bs) with their 116C-NOD transgenic counterparts were cultured under different conditions including: without stimulus (ns), with lipopolysaccharide (LPS), with anti-B cell receptor (α BCR), and with anti-CD40 (α CD40) plus IL-4 (n=4 for each culture condition). IL-4 ctrl: control wells without cells and with the same IL-4 concentration (to assess IL-4 consumption by B cells). Data are expressed as mean \pm SD.

In relation to this issue, we have corrected the legend of Fig. 2 (lines 878-879):
 Fig. 2 | *In vitro* cytokine secretion analysis of T and B lymphocytes from NOD mice isolated and cohoused with 116C-NOD mice.

In response to the second part of the question regarding lymphocyte proliferation:
 We conducted proliferation assays within the same *in vitro* cell culture experiments (co-cultures of T and B cells, as well as cultures of B cells) (Supplementary Fig. 4). No differences were found between isoNOD T cells and coNOD T cells. T cell proliferation was not influenced by isoNOD B cells nor coNOD B cells. Moreover, no differences were found between isoNOD B cells and coNOD B cells.

Supplementary Fig. 4 | Proliferation index of CD4⁺ and CD8⁺ T cells, and B cells from NOD mice isolated and cohoused with 116C-NOD mice. T cells from NOD mice isolated (isoNOD Ts) and cohoused (coNOD Ts) were cultured *in vitro* under different conditions: alone, with well-coated or fixed anti-CD3 (F α CD3), in the presence of soluble anti-CD3 (α sCD3), and co-cultured with B cells from NOD mice isolated (isoNOD Bs) and cohoused (coNOD Bs), in their four possible combinations, plus α sCD3 (n=4 for each culture condition). Data are expressed as mean \pm SD and analysed with the Mann-Whitney and two-tailed t-test.

There is no directly *ex vivo* immunology data from the mice studied, only co-cultures, it would be good to study the T and B cell phenotyping in more detail in different tissues e.g. gut, pancreas, pancreatic lymph nodes to show *in vivo* evidence microbiota/housing directly effects the immune cells.

We appreciate the reviewer's suggestion and as a result, we have incorporated an experiment involving direct *ex vivo* phenotyping of T and B cell subsets. These subsets were analyzed in the spleen, mesenteric lymph nodes (MLN), Peyer's patches (PP), cecal patch (CP), and pancreatic islet infiltrate of isolated and cohoused NOD mice. The evaluated CD4⁺ and CD8⁺ T cell subsets encompassed the following: naïve T cells, effector T cells, effector memory T cells, central memory T cells, tissue-resident memory T cells, exhausted T cells, and anergic T cells. The assessed B cell subsets included: follicular B cells, marginal zone B cells, transitional B cells, germinal center B cells, memory B cells, plasmablasts and plasmacytes. Please refer to the specified lines in the Methods section for further details (lines 713-767).

Our findings indicated that the natural transfer of gut microbiota via cohousing influences the composition of T and B cell subsets of MLN, gut-associated lymphoid tissue (PP and CP), and pancreatic islet infiltrate. For more specific information, refer to the corresponding sections in the Results (lines 190-210), along with Fig. 4 and Supplementary Fig. 5.

While the authors identify markers of prediction, the authors did not use these microbial markers to predict diabetes development – others have done this previously (<https://www.nature.com/articles/s41598-018-33571-z>) so this should be conducted to confirm their claim.

Due to the size of our cohort, we consider that following an approach involving the prediction of diabetes is inappropriate. The authors in the mentioned paper formulate a predictive algorithm (logistic regression) for T1D development using 63 NOD mice as the training or discovery cohort while using 29 mice as the test cohort.

Our study cohort consists of 46 NOD mice, which have to be furtherly separated between co-housing (21 mice) and isolated (25 mice). Due to the inavailability of an independent test set of mice with these conditions, we would be forced to split our dataset into training and test cohort. Using the same design as Hu et al., the paper mentioned by the reviewer; train set ≈ 54%, and test set ≈ 46%; the cohort would result as follows:

- NOD-coh: 11 mice train set & 10 mice test set
- NOD-iso: 14 mice train set & 11 mice test set

This approach would likely result in poor performance of the model due to the small size of the training set.

On the other hand, if we follow a more conservative approach, using 2/3 of the cohort as the training set and 1/3 as the test set, the cohort would result as follows:

- NOD-coh: 14 mice train set & 7 mice test set
- NOD-iso: 17 mice train set & 8 mice test set

This scenario does not improve the size of the training cohort, while it decreases significantly the size of the test cohort, making it impossible to extrapolate the results of such a model to the population reliably.

To illustrate the mentioned points, we performed logistic regression in the best possible scenario (sample-wise), where we predicted T1D development using a general marker for all mice types and housing conditions (Clostridium genus at week 20). In this case, the cohort

consists of 37 mice (11 NOD-coh, 14 NOD-iso and 12 116NOD-iso), which leads to a training set of 26 mice and a test set of 11 mice.

This logistic regression model achieved 54.54% accuracy, correctly classifying 6 out of 11 samples between future diabetic and future resistant. ROC analysis, used to test the sensitivity and specificity of the parameters used for diabetes prediction, outputted an area under the curve (AUC) of 0.567. Upon closer observation of the curve, it is obvious that the number of samples is not enough to rely on the predictive power of the model.

(insert 0.57 ROC AUC plot here)

The poor reliability of the prediction due to the sample size is even more clear if we construct different training and tests set from the same samples (still a training set of 26 mice and a test set of 11 mice, but composed of different mice).

In this case, the logistic regression model achieves 72.72% accuracy, correctly classifying 8 out of 11 samples between future diabetic and future resistant. In this case, ROC analysis outputs an area under the curve (AUC) of 0.8.

In these analyses, we observe a dramatic change in the predictive power of the model by only resampling the training and test sets, which should not happen if the sample size was big enough to account for mice variability.

It is unclear how specifically the microbiota influence immune cells and how immune cells change microbiota. Investigation of mice treated with antibiotics, evaluating intestinal permeability etc may help provide more idea as to mechanism of action.

We appreciate the reviewer's suggestion, and in response, we conducted an experiment to assess intestinal permeability using FITC-dextran assay. This experiment involved isolated NOD, cohoused NOD, isolated 116C-NOD, NOD.RAG-2^{-/-}, 116C-NOD.RAG-2^{-/-} and C57BL/6J mice. Please refer to the specified lines in the Methods section for further details (lines 812-831).

Our findings demonstrate that intestinal permeability is influenced both by gut microbiota and the immune system. Specifically, isolated NOD mice exhibited significantly higher levels of gut permeability when compared to cohoused NOD and isolated 116C-NOD mice. In addition, the intestinal permeability of 116C-NOD.RAG-2^{-/-} mice was reduced compared to NOD.RAG-2^{-/-}. However, no differences were observed between 116C-NOD and 116C-NOD.RAG-2^{-/-} mice. Please refer to the corresponding section of Results (lines 394-406).

Therefore, our findings suggest that alterations in gut permeability may serve as a potential underlying mechanism for the observed effects of the gut microbiota and the immune system (please refer to lines 524-544 of the Discussion section).

Cohousing is the approach we employed to investigate the modulation of gut microbiota. Other alternative approaches include fecal matter transfer and antibiotic administration, among others. Regarding the use of antibiotics in animal models of T1D, a study conducted by the laboratory of Prof. Martin Blaser provided valuable insights into the interplay between gut microbiota and the immune system. They observed that the administration of the macrolide antibiotic tylosin tartrate, which had significant effects on the composition of the gut microbiota, altered the expression of genes involved in immunological functions such as cell adhesion, T cell receptor signalling, and B cell receptor signalling (Zhang *et al.*, 2021, <https://pubmed.ncbi.nlm.nih.gov/34289377/>).

The lymphocyte transfers were conducted at a late age (6 weeks of age) when most Rag mice receiving transferred cells may not develop diabetes – did the authors observe diabetes incidences for the rag transfer studies and did this support the microbiota associations with diabetes development?

Firstly, in previous unpublished research conducted within our laboratory, we observed that 10-weeks-old NOD.RAG-2^{-/-} females, transferred with splenocytes from NOD females of the same age, could develop autoimmune diabetes (please see figure below). Additionally, in a previous study, Söderström *et al.* found similar outcomes by transferring splenocytes from NOD females to NOD.RAG-2^{-/-} at 6 weeks of age (<https://pubmed.ncbi.nlm.nih.gov/8633210/>). Therefore, it can be concluded that NOD.RAG-2^{-/-} mice that received NOD splenocytes at 6 weeks of age or later have the potential to develop diabetes.

Secondly, in the current study, none of the NOD.RAG-2^{-/-} mice transferred at 6 weeks with splenocytes, T cells or B cells from 6-weeks-old NOD mice developed autoimmune diabetes at the time of sacrifice (12 weeks of age). These experiments were conducted to analyse the effect of the T and B lymphocyte repertoire on the composition of gut microbiota. For this reason, T1D monitoring beyond 12 weeks of age was not carried out.

Regarding the mice that were transferred with NOD splenocytes, no instances of diabetes were detected, as they were sacrificed at 12 weeks of age, making it impossible to track the disease beyond this time point. The same situation occurred for mice transferred with T cells. It is worth noting that transgenic 4.1-NOD.RAG-2^{-/-} mice, which only bear beta cell-specific CD4⁺ T cells, develop diabetes as early and as frequently as 4.1-NOD.RAG-2⁺ mice (please see Verdaguer *et al.*, 1997, <https://pubmed.ncbi.nlm.nih.gov/9362527/> (reference number 8 of the manuscript)). Hence, it is expected that the incidence of autoimmune diabetes in NOD.RAG-2^{-/-} transferred with NOD T cells is would be similar to that of mice transferred with NOD splenocytes. Regarding the mice that received purified B cells, they could not become diabetic since it is known that, in the absence of T cells, B cells cannot induce autoimmune diabetes (Bendelac *et al.*, 1987: <https://pubmed.ncbi.nlm.nih.gov/3309126/>, Koike *et al.*, 1987: <https://pubmed.ncbi.nlm.nih.gov/3102302/>).

From Figure 4 onwards– the text seems to jump around from one figure to the next then back to the former, which got quite confusing when reading. My suggestion would be to move the figures around in the order presented in the text for easier reading.

We appreciate and understand the reviewer’s suggestion. We would prefer to merge current figures 5 and 6a (previous figures 4 and 5a). However, due to the type of graphs and the size restriction of figures, we decided to classify the graphs into two groups: future-diabetics and future-resistants graphs (Fig. 5) (the biggest graphs, all of them sharing the same tipology), and the remaining graphs in Fig. 6.

Further methodological information should be provided e.g. Were mice studied housed in the same room? Were Rag2^{-/-} mice treated with antibiotics? How many times were the experiments repeated and were the data reproducible?

All the cages were located in the same SPF room. NOD.RAG-2^{-/-} were not treated with antibiotics. The rest of the models were not treated with antibiotics either. Immunological studies were successfully repeated twice.

Minor issues:

Islet infiltration – early not different but what about 40 weeks at the end of observation?
Unfortunately, we do not have frozen pancreas of 40 weeks available to study islet infiltration. Nevertheless, as mentioned before in the response to the second question, we evaluated the composition of the islet infiltrate in terms of T and B cell subsets at 12 weeks of age. Higher percentages of effector CD4⁺ and CD8⁺ T cells were observed in the islet infiltrate of isolated NOD mice. On the contrary, anergic CD4⁺ T cells were found in higher percentages in pancreatic islets of cohoused NOD mice. These results indicated that the composition of the islet infiltrate, rather than the degree of infiltration, was influenced by cohousing conditions.

Figure 1 – make it easier to see the 0 incidence rates by separating the 2 lines on the x axis
We have raised the line of 116C-NOD.RAG-2^{-/-} incidence above the line of NOD.RAG-2^{-/-} incidence.

Figure legend 1 – T is missing from T1D on line 668.
We have corrected the first word of this legend (line 858).

Line 158 – I am not sure TNF α produced by T cells was highly modulated by the presence of co-B cells as TNF α as the T cells have a bigger effect housed separately and only a small difference is seen between iso T cells +CoBs or isoTs – suggest removing this sentence
We have removed TNF-alpha from the cytokine secretion analysis of T cells (Fig. 2a), in relation to the first question of the reviewer.

Line 188 – GATA3 does show a trend but I would remove mention of FoxP3 as that does not appear to.
We corrected the sentence (lines 184-185):
“Lastly, the CD4⁺ GATA3⁺ T lymphocyte subpopulations showed an upward trend in coNOD Ts compared with isoNOD Ts.”

Reviewer #3 (expert in gut microbiota and sequencing):

In this manuscript, Rosell-Mases et present the potential of the microbiota in the development of T1D as well as explore its interest as a predictive target.
The value of the work is of undoubted interest for public health as the authors conclude, as these findings open the door to personalized therapies in the next-future. While I do not doubt the high quality of the scientific work and hard work behind, I do believe that it would be fundamental that the authors look deep into the mechanisms by which the gut microbiota shape the T1D risk starting from these preclinical observations. In the current version, the analyses of the gut microbiota are restricted to 16S RRNA gene sequencing (metabolomics and metagenomics would be ideal) and there are no analyses that tackle the host-microbiota cross-talk. For instance, does the gut microbiota of NOD cohoused with 116C-NOD influence on the intestinal immune with consequences in the autoimmune response? If so, which bacterial metabolites rule are mediators? Or maybe any structural compound from bifidobacterial strains?

We have incorporated an experiment involving direct *ex vivo* phenotyping of T and B cell subsets. These subsets were analyzed in the spleen, mesenteric lymph nodes (MLN), Peyer's patches (PP), cecal patch (CP), and pancreatic islet infiltrate of isolated and cohoused NOD mice. The evaluated CD4⁺ and CD8⁺ T cell subsets encompassed the following: naïve T cells, effector T cells, effector memory T cells, central memory T cells, tissue-resident memory T cells, exhausted T cells, and anergic T cells. The assessed B cell subsets included: follicular

B cells, marginal zone B cells, transitional B cells, germinal center B cells, memory B cells, plasmablasts and plasmacytes. Please refer to the specified lines in the Methods section for further details (lines 713-767).

Our findings indicated that the natural transfer of gut microbiota via cohousing influences the composition of T and B cell subsets of MLN, gut-associated lymphoid tissue (PP and CP), and pancreatic islet infiltrate. For more specific information, refer to the corresponding section in the Results (lines 190-210).

Furthermore, we conducted an experiment to assess intestinal permeability using FITC-dextran assay. This experiment involved isolated NOD, cohoused NOD, isolated 116C-NOD, NOD.RAG-2^{-/-}, 116C-NOD.RAG-2^{-/-} and C57BL/6J mice. Please refer to the specified lines in the Methods section for further details (lines 812-831).

Our findings demonstrate that intestinal permeability is influenced both by gut microbiota and the immune system. Specifically, cohoused NOD mice exhibited significantly lower levels of gut permeability when compared to isolated NOD and 116C-NOD mice. In addition, the intestinal permeability of 116C-NOD.RAG-2^{-/-} mice was reduced compared to NOD.RAG-2^{-/-}. However, no differences were observed between 116C-NOD and 116C-NOD.RAG-2^{-/-} mice. Please refer to the corresponding section of Results (lines 394-406).

Therefore, our findings suggest that alterations in gut permeability may serve as a potential underlying mechanism for the observed effects of the gut microbiota and the immune system (please refer to lines 524-544 of the Discussion section).

Mayor comments:

1. Authors should justify the decision of considering FDR lower than 0,10 as significant.

Response: We agree that the choice of FDR threshold should be carefully justified, and it should be considered in the context of the specific research question, experimental design, and the characteristics of the microbiome data being analyzed.

The use of a False Discovery Rate (FDR) threshold of less than 0.1 instead of less than 0.05 for microbiome analysis is based on common scientific practices and considerations. Indeed, using a less stringent FDR threshold, such as 0.1, can increase the statistical power of the analysis, allowing for the detection of potentially important microbiome differences that may be missed with a more conservative threshold like 0.05. This can be especially relevant in microbiome research, where sample sizes are often limited. Moreover, it allows us to generate hypotheses for further investigation rather than making definitive conclusions. Using a higher FDR threshold of 0.1 can be justifiable in such scenarios, as it allows for a more liberal approach to identifying potentially interesting patterns or trends in the microbiome data. Microbiome data can be highly complex, with multiple taxa or features being tested simultaneously, and traditional significance thresholds like 0.05 may be overly conservative.

2. Authors refer to α -diversity, based on Chao1, which is more "richness". It is true that sometimes authors refer to Shannon (diversity index), but not always, like in lines 440-441, where they claim lower or higher diversity, when only the Chao1 index changes, but the Shannon index is the same.

We appreciate the reviewer's comment and agree that Chao1, which is a non-parametric estimator of the number of species in a population, reflects the microbial richness, as it takes into account both the number of observed species and the number of singletons (species with only one individual observed). Whereas the Shannon index is a measure of the diversity and evenness of a community, which takes into account both the number of species present and the relative abundance of each species.

In our study, we calculated both Chao1 and Shannon indexes for alpha diversity analysis and reported the indexes only when they were significantly different between groups. We will now make the correction in the manuscript to replace diversity with richness when appropriate.

3. Why the gut microbiota of cohoused 116C-NOD was not analyzed (Fig4)? How does the author explain the absence of differences between NOD and 116C-NOD (Fig 6)?

The analysis of the gut microbiota in cohoused 116C-NOD mice (n=78 samples) was not conducted, due to financial constraints and the sufficient information obtained from analyzing the gut microbiota of cohoused NOD mice. Indeed, for this project, a total of 360 samples have already been analysed, which was far over our initial budget.

NOD and 116C-NOD mice evidenced significant differences in the relative abundance of *Bifidobacterium* (current figure 7, previous figure 6 which the reviewer referred to). Please see lines 347-348: "... among the T1D-prone animals, the 116C-NOD model had higher levels of *Bifidobacterium* compared with isolated and cohoused NOD mice ($q=7 \times 10^{-5}$) (Fig. 7a)."

Moreover, in figures 5 and 6 (previously referred to as figures 4 and 5), differences between isoNOD and iso116C-NOD are illustrated. We have added the following information (please see lines 297-303):

"Isolated NOD mice and 116C-NOD presented a different pattern of gut microbiota in terms of the Rikenella genus and the Actinobacteria and Deferribacteres phyla. Related samples of future-diabetic mice at 6, 12, and 20 weeks of age displayed a lower relative abundance of Rikenella in isolated NOD mice ($q=0.088$) (Fig. 5). Additionally, Actinobacteria were more represented in 116C-NOD mice at six weeks of age ($q=0.052$) (Fig. 6b). Interestingly, the gut microbiota of diabetic animals from the isolated NOD group was enriched in Deferribacteres when compared to 116C-NOD ($q=0.054$) (Fig. 6a)."

4. Authors described that CoNOD has low levels of Sutterella and Proteobacteria acquired from cohousing as a protective microbe against T1D (393-401). However, these genera have the same prevalence in isoNOD and iso116C-NOD. How do the authors justify this claim?

The reviewer's question highlights the complexity of the gut microbiome community and its interaction with the host immune system. Sometimes, the role or impact of a single microbial genus (singular form of genera) can be straightforward and easily explained. However, in other cases, understanding the dynamics and effects may require considering the interactions and complexities of a complex microbial network. This statement acknowledges the varying levels of complexity involved in studying the gut microbiome and its relationship with the host immune system.

5. Regarding the use of the microbiota as a possible predictive marker. Authors compare at 12 weeks when they are already starting to develop diabetes, and not at 6 weeks. However, I think it would be interesting to compare also at 6 weeks as this time point might pinpoint better mice more "prone" to develop diabetes.

In the initial version of the manuscript, we did analyse the gut microbiota as a marker at 6 weeks of age. Please see:

- Lines 218-221: "...At six, 12, and 20 weeks of age, isolated NOD, cohoused NOD, and isolated 116C-NOD mice were divided into two groups: mice that became diabetic over

the 40 weeks of disease follow-up (future diabetics) and mice that remained resistant until the end of this period (future resistants).”

- **Lines 241-247:** “...future-diabetic isolated 116C-NOD mice exhibited an increased relative abundance of the phylum Tenericutes ($q=0.049$) at **six** weeks of age compared with future-resistant mice and also when comparing correlated samples of **six**, 12, and 20 weeks of future diabetics with future-resistant mice ($q=0.093$). Furthermore, the unknown genus of Cyanobacteria order YS2 was enriched in correlated samples of **six** and 12-week-old future-diabetic mice ($q=0.052$). In parallel, the unknown genus of the Alphaproteobacteria order RF32 was enriched in related samples of **six**, 12-, and 20-week-old future-diabetic mice ($q=0.043$) (Fig. 5).”

5. Do they confirm the differences found *in vitro* *in vivo* in the cytokine levels?

We appreciate the reviewer’s suggestion.

Most cytokines are utilized, catabolized, or excreted shortly after their production (Finkelman *et al.*, 1993, <https://pubmed.ncbi.nlm.nih.gov/8393043/>). In addition, its secretion is frequently local and not systemic with an autocrine and/or paracrine effect. Only in highly inflammatory processes such as toxic shock syndrome, sepsis, endotoxin shock or severe viral infections such as Sars-Cov-2, characterized by a very high production of cytokines (cytokine storm), it is possible and easy to measure in serum or plasma. On the contrary, in cases of chronic and highly localized diseases, such as the *in situ* response in the pancreatic islets in autoimmune diabetes, the production of cytokines *in vivo* is difficult to detect. Moreover, measuring systemic cytokines produced by B cells can be challenging due to several other factors, such as:

- Low abundance: B cell-derived cytokines are often present in low quantities compared to cytokines produced by other cell types. This can make their detection and quantification more difficult. (Vazquez, 2015, doi: 10.1016/j.cyto.2015.02.007).
- Cell-specificity: Cytokines produced by B cells can also be produced by other cell types, making it challenging to specifically attribute their production to B cells. It requires sophisticated techniques to differentiate B cell-derived cytokines from those produced by other cells.
- Context dependency: The production of B cell-derived cytokines can be influenced by various factors, including the microenvironment, co-stimulatory signals, and interactions with other immune cells. The systemic measurement of B cell-derived cytokines may not always reflect their local production in specific tissues or organs.

Only the use of more complex techniques such as the *in vivo* administration of cytokine-specific neutralizing antibodies that selectively capture cytokines *in vivo* before their degradation, to later analyze their concentration, can partially alleviate this deficiency (Current Protocols in Immunology, Unit 6.28). This technology is complex and difficult to apply systematically to analyze a large number of cytokines in a large number of individuals. For this reason, researchers usually test cytokine production *in vitro*. This allows analysis of the cytokine production of the desired population without difficulty and with high reliability. Thus, in the present study, we analyzed cytokine secretion by T and B lymphocytes *in vitro* and did not perform cytokine analysis *in vivo*.

6- Can the authors provide any regarding the soundness of their hypothesis in human data?

We have improved the discussion regarding the hypothesis in human data, as follows:

Cohoused NOD mice presented a reduction in *Sutterella* and *Proteobacteria*. In line with this finding, Li Wen's team also found a decrease in *Sutterella* and *Proteobacteria* in TRIF-deficient mice, which were protected against T1D development, compared with wild-type NOD. In humans, *Proteobacteria* was increased in biopsy samples from Crohn's disease patients and *Sutterella* was also increased in the inflamed mucosa of patients with inflammatory bowel disease²⁵ (lines 460-468).

Regarding the Th1/Th17 shift found in cohoused NOD mice, this double-sided Th response was observed in prediabetic children with advanced beta cell autoimmunity, but not in overt T1D²² (lines 437-439).

We observed that segmented filamentous bacteria (SFB) were present in NOD and 116C-NOD mice at six weeks of age, but gradually decline and disappeared completely by 20 weeks. However, immunodeficient NOD.RAG-2^{-/-} and 116C-NOD.RAG-2^{-/-} mice continue to have SFB throughout their lifespan. In contrast SFB was not detected in non-T1D-prone C57BL/6J mice at any age. In humans, SFB colonisation varied with age. SFB was present in human faecal samples until 36 months of age, with the highest occurrence (78.6%) observed between seven and 12 months. Fewer adults between 21 and 41 years (30.0%) and 41 and 51 years (16.7%) had SFB in their gut microbiota²⁶. Hence, this suggests that young NOD and 116C-NOD mice may resemble infants colonised with SFB, and these children may have lymphocyte development imbalances that are resolved as the immune system matures. Similarly, C57BL/6J mice could be equivalent to children without SFB, displaying a balanced lymphocyte development (lines 488-495).

Furthermore, our findings indicated that the colonisation of *Bifidobacterium* required the presence of lymphocytes, and were particularly enhanced in a non-diabetogenic milieu. Our results align with numerous studies that have reported a decrease of *Bifidobacterium* within the gut microbiota of T1D patients²⁷⁻³⁰ (lines 496-499).

Interestingly, we observed that 116C B cells could specifically induce the colonisation of bacteria such as *Adlercreutzia* and *Parabacteroides*. Interestingly, both genera were less abundant in the gut microbiota of multiple sclerosis patients³¹ (lines 516-519).

Finally, we uncovered a connection between the gut microbiota and the immune system in their impact on intestinal permeability. Isolated NOD mice displayed higher levels of gut permeability compared to cohoused NOD and isolated 116C-NOD mice. Moreover, 116C-NOD.RAG-2^{-/-} mice exhibited lower levels of gut permeability compared to NOD.RAG-2^{-/-}, but not to 116C-NOD mice. This suggests that intestinal permeability represents a potential underlying mechanism for the observed effects of the gut microbiota and the immune system. In humans, increased gut permeability has also been associated with T1D³³⁻³⁴ (lines 524-540).

REVIEWER COMMENTS

Reviewer #1 (expert in murine models of type-1diabetes and gut microbiome):

The authors have made amendments to the manuscript related to the critical points. The new figure (Supplementary Fig. 2) now provides progressive diabetes survival curves of mice chosen on the basis of high - or low abundance of selected taxa (Cyanobacteria YS2, (Alpha)proteobacteria, Tenericutes and Clostridium) as well as microbiota richness (Chao1).

The survival curves confirm robust associations between diabetes incidence and abundance (hi-lo) of the reported taxa in gut microbiota. However, the concern still remains that the reported abundances of all identified taxa are very low, typically between 0.001 - 0.004. As the authors comment, taxonomic annotation of 16SRNA amplicon sequencing data is subject to database and primer selection, but the concern still exists, that significant shifts in microbiota composition remain veiled.

The authors have added a sentence to the Discussion commenting the impact of co-housing already weaned mice on efficacy of microbiota modulation.

Reviewer #2

Absent but replaced by reviewer 4.

Reviewer #3

Absent but replaced by reviewer 5.

Reviewer #4 (expert in immunology of T1D):

Many concerns from reviewer 2 have been addressed. Still, key questions are pending.

Point 1 (related to Figure 2): I understand that most cytokines are either produced by T cells or B cells. While the method to infer that cytokines are produced by T cells is acceptable, what about IL-6 production? Especially as the authors claim to focus on IL-6 production.

Point 2 (No direct ex vivo data): The experiment is helpful and answer the reviewers point. However, it seems that the experiment as only been performed once, with a total of 3 mice per group. This is very low for a characterization with such high level of variability. More mice should be analyzed, in a minimum of 3 independent experiment. The statistical analysis also does not seem appropriate. The data has probably been tested for significance with a T test more than a Man Whitey test (both are listed in the figure legend). T test used to analyze data with such low repeat number and high variance are not reliable.

Point 3 (prediction): Ok

Point 4 (influence of microbiota): This point is not sufficiently addressed. While the authors show an interesting impact of co-housing on gut permeability, that does not demonstrate how it impacts immune cells and their functions. Identification of the mechanism explaining the link between microbiota and T1D incidence changes would greatly improve this manuscript.

Point 5 (lymphocyte transfer): The authors state: "Hence, it is expected that the incidence of autoimmune diabetes in NOD.RAG-2-/- transferred with NOD T cells is would be similar to that of mice transferred with NOD splenocytes".

This interpretation is over simplified. Multiple groups have shown that presence of B cells (which would be present in total spleen cell transfer) have an impact on diabetes incidence in transfer models.

Ex: doi: 10.1172/jci.insight.99860, doi: 10.4049/jimmunol.1700024

The authors do not sufficiently address the reviewer's concern.

Point 6 (figure and text readability past figure 4): This remains an issue. The description of Figures 5 and 6 is not ordered which makes it difficult to read and to follow the train of thought.

Point 7 (methodological information): The number or repeated independent experiments should be clarified in each figure legends. Each experiment should be tested a minimum of three independent repeats.

Additional concerns on new data and text:
Specific comments:

Figure S2: I do not understand what is plotted on the Y axis? What is a future diabetes incidence? How could the % of future diabetic mice change over time since it represents the % of mouse that will be diabetic in the future. Or is it the % of diabetic mice? If so, please correct the Y axis title everywhere. Also, description of the data from this figure in the main text is long and unnecessary. It could be shortened.

Line 335 and 338: NOD.Rag^{-/-} mice also lack B cells and 116-C.NOD.Rag^{-/-} mice lack a polyclonal B cell population (therefore B cells that could recognize microbial antigens). B cells could therefore be at cause, especially as the transfer of NOD B cells also decreased the presence of filamentous bacteria (Fig. 7b).

Line 477: Similarly, the authors should write "T cells disorders are a driver" rather than "the driver".

Line 349: "The presence of lymphocytes is a requirement for Bifidobacterium colonization". Is it possible that some strains (B6, NOD, etc) are simply colonized in the animal facility while others are not (NOD.Rag^{-/-} etc). To prove that the presence of lymphocytes is a requirement would require co-housing of NOD (or B6) with NOD.Rag^{-/-} mice to see if colonization can take place. Same thing for the claim made in line 361/362.

Line 372: 7a should be 7c

Reviewer #5 (expertise microbiota and T1D):

The authors have now added new data showing the impact on intestinal permeability and immune populations in the GALT tissues following co-housing with 116C-NOD mice. This new data improves the manuscript by adding extra evidence for how the 116C microbiota may be impacting the host. However, the specific microbiota pathways that are mediating these effects are still unknown and knowledge of this is limited by only using 16S amplicon profiling of the microbiota not metagenomics or metabolomics (e.g. SCFA measurement or more comprehensive metabolite profiling). Therefore, this remains a significant limitation that needs to be acknowledged in the discussion.

4. The lack of direct functional microbiome profiling means it is hard to explain some of the bacterial changes which are both lower in protected coNOD mice but higher in iso116C-NOD mice (*Sutterella* and *Proteobacteria*). Furthermore, these two taxa were not predictive of future diabetes in the new Suppl Fig 2. Therefore the relevance of changes in these taxa is unclear and the paragraph in the discussion (lines 458-466) on these two taxa should be removed.

5. Cytokine levels were not checked in vivo – this could have been done by FACS with Intracellular cytokine staining ex vivo or via IHC or IF staining on tissue sections. This also should be acknowledged as a limitation in the discussion.

6. Lines 514-517: It is noted that *Parabacteroides* is lower in multiple sclerosis patients. Yet, *Parabacteroides* was found to be higher in children that progressed to type 1 diabetes (Stewart et al

Nature 2018: Temporal development of the gut microbiome in early childhood from the TEDDY study). Parabacteroides was also increased in adults with T1D after consuming a resistant starch diet that is associated with diabetes protection in mice (Bell et al 2022 Microbiome). Therefore, it remains unclear whether Parabacteroides has a beneficial or detrimental role in T1D. The discussion should be updated to reflect this.

Minor point: Lines 199-200 T cell populations are labelled as exhausted and anergic based on surface expression of PD-1 and Lag-3 or CD73 and FR4. Yet more conclusive evidence (e.g. functional or confirming expression of more markers such as EOMES) would be needed to definitively define these populations as such. Therefore, the populations should be referred to as 'exhausted-like' and anergic-like' cells. It would also be helpful for the reader to include the markers gated for the immune populations in the Figure (e.g. in the axis labels or in the legend).

Dear Reviewers,

We thank you for the time spent reviewing our manuscript and appreciate the opportunity to address your comments and criticisms. In this regard, we feel that all the points you raised have allowed us to improve the overall quality and readability of the manuscript. Our answers are written in blue below your comments. The modifications made to the manuscript have been added with track changes.

We hope that you now find the new version suitable for publication.

REVIEWER COMMENTS

Reviewer #1 (expert in murine models of type-1diabetes and gut microbiome):

The authors have made amendments to the manuscript related to the critical points. The new figure (Supplementary Fig. 2) now provides progressive diabetes survival curves of mice chosen on the basis of high - or low abundance of selected taxa (Cyanobacteria YS2, (Alpha)proteobacteria, Tenericutes and Clostridium) as well as microbiota richness (Chao1).

The survival curves confirm robust associations between diabetes incidence and abundance (hi-lo) of the reported taxa in gut microbiota. However, the concern still remains that the reported abundances of all identified taxa are very low, typically between 0.001 - 0.004. As the authors comment, taxonomic annotation of 16SRNA amplicon sequencing data is subject to database and primer selection, but the concern still exists, that significant shifts in microbiota composition remain veiled.

Response: We acknowledge the reviewer's comment. Indeed, it is challenging to envision that the changes occurring in low-abundance taxa could have a significant impact on modulating the immune system. New experiments could be designed to assess the effect of these low-abundance taxa on immunogenicity. While conducting a literature search, we came across an interesting paper that proposes such an experiment (doi: 10.1080/19490976.2022.2104086). In this study, the authors employed a microbiome dilution strategy to reduce the abundance of low-abundance bacteria selectively and then transplanted the diluted microbiomes into germ-free mice. They observed that the depletion of low-abundance bacteria led to a dramatic reduction in the expression of multiple genes involved in the MHCII antigen presentation pathway and T-cell cytokine production in the small intestine. This study highlights that even taxa present in low abundance (less than 1%) can exert a significantly higher immunostimulatory effect than dominant bacteria.

The authors have added a sentence to the Discussion commenting the impact of co-housing already weaned mice on efficacy of microbiota modulation.

Reviewer #2

Absent but replaced by reviewer 4.

Reviewer #3

Absent but replaced by reviewer 5.

Reviewer #4 (expert in immunology of T1D):

Many concerns from reviewer 2 have been addressed. Still, key questions are pending.

Point 1 (related to Figure 2): I understand that most cytokines are either produced by T cells or B cells. While the method to infer that cytokines are produced by T cells is acceptable, what about IL-6 production? Especially as the authors claim to focus on IL-6 production.

Response: We acknowledge the reviewer's comment. There was indeed a missing piece of text in the answer to reviewer 2, which was accidentally deleted during the final document editing process. We have now included the missing information, highlighted in yellow and underlined as follows:

"...In the B cell *in vitro* culture (Fig. 2b), we observe the secretion of IL-6 and IL-10 by B lymphocytes upon stimulation with LPS. However, when stimulated with anti-BCR or anti-CD40 plus IL-4, IL-6 and IL-10 were not produced (very low levels of IL-10 in anti-CD40 plus IL-4 condition). Consequently, it can be inferred that T cells are responsible for the production of IL-6 and IL-10 in the co-culture, as illustrated in Fig. 2a..."

Point 2 (No direct ex vivo data): The experiment is helpful and answer the reviewers point. However, it seems that the experiment as only been performed once, with a total of 3 mice per group. This is very low for a characterization with such high level of variability. More mice should be analysed, in a minimum of 3 independent experiment. The statistical analysis also does not seem appropriate. The data has probably been tested for significance with a T test more than a Man Whitey test (both are listed in the figure legend). T test used to analyse data with such low repeat number and high variance are not reliable.

Response: We agree with the reviewer that three mice per group are very low. The experiment was duplicated, as requested by the editor herself. For the analysis of both independent experiments, we used two-way ANOVA tests on rank-transformed data-values to avoid violating normality and homoscedasticity assumptions of this parametric test, as recommended by the head of our statistical department and described now in the legends of our figures, and not the t-test (text now corrected). This new statistical analysis methodology has been included in the manuscript (please see response to point 7).

Point 3 (prediction): OK

Point 4 (influence of microbiota): This point is not sufficiently addressed. While the authors show an interesting impact of co-housing on gut permeability, that does not demonstrate how it impacts immune cells and their functions. Identification of the mechanism explaining the link between microbiota and T1D incidence changes would greatly improve this manuscript.

Response: Conducting co-housing and cell transfer experiments has provided conclusive evidence of the gut microbiota's direct involvement in the development of T1D. However, pinpointing the precise mechanisms responsible for the fluctuations in T1D incidence would necessitate the formulation of novel experiments. Additionally, identifying these mechanisms would utilize advanced techniques such as shotgun DNA/RNA metagenomics or metabolomics approaches, followed by the transplantation of the identified microorganisms or the transfer of microbial products into an animal model. In response to the reviewer's suggestion and also as

recommended by reviewer #5, we are in the process of designing a new project specifically aimed at addressing this question. We added this limitation in the discussion section as follows:

“...To gain a deeper understanding of the specific pathways and mechanisms connecting gut microbiota to T1D, advanced techniques of microbiota analysis, such as shotgun DNA/RNA metagenomics or metabolomics approaches, along with direct *ex vivo* cytokine profiling analyses, are necessary...”

Point 5 (lymphocyte transfer): The authors state: "Hence, it is expected that the incidence of autoimmune diabetes in NOD.RAG-2^{-/-} transferred with NOD T cells is would be similar to that of mice transferred with NOD splenocytes". This interpretation is over simplified. Multiple groups have shown that presence of B cells (which would be present in total spleen cell transfer) have an impact on diabetes incidence in transfer models. Ex: doi: 10.1172/jci.insight.99860, doi: 10.4049/jimmunol.1700024

The authors do not sufficiently address the reviewer's concern.

Response: The reviewer's observation is indeed accurate. It is well-established that B lymphocytes with regulatory activity exist. We would like to emphasize that the transfer of splenocytes and autoreactive T lymphocytes from NOD mice to immunodeficient NOD.SCID and NOD.RAG-2^{-/-} mice usually results in a high cumulative incidence of autoimmune diabetes in both cases, as has been demonstrated in multiple studies (doi: 10.2337/diab.45.3.328. doi: 10.2337/diab.44.5.550; doi: 10.2337/diab.42.1.44; doi: 10.1007/s001250050786. doi: 10.1084/jem.183.1.67). Therefore, a more appropriate statement would be: “Therefore, the incidence of autoimmune diabetes in NOD.RAG-2^{-/-} transferred with NOD T cells is also expected to be high.”

Point 6 (figure and text readability past figure 4): This remains an issue. The description of Figures 5 and 6 is not ordered which makes it difficult to read and to follow the train of thought.

Response: We acknowledge the reviewer's comment. We have now improved the readability of the text, as now referred to the figures 5, 6, and 7 (previous figures 5, 6a, and 6b).

Point 7 (methodological information): The number or repeated independent experiments should be clarified in each figure legends. Each experiment should be tested a minimum of three independent repeats.

Response: We appreciate the reviewer's comment and concur with the necessity to add this information to each figure legend. Each experiment underwent two independent repeats (as requested by the journal editor). Both repeats were plotted in the graphics and analysed with two-way ANOVA tests on rank-transformed data-values. We added the information related to the statistical analysis in the Methods section as follows:

“...The cytokine profiling, transcription factor analysis, proliferation assay, direct *ex vivo* immunophenotyping of lymphocyte subsets, and intestinal permeability analysis were conducted in two independent experiments (the total number of samples, including both experiments, is shown in each graphic and figure legend). To assess reproducibility of the experiments, we performed two-way ANOVA tests on rank-transformed data-values to avoid violating normality and homoscedasticity assumptions of this parametric test; using mice/condition as one of the groups, and experiment (repetition 1 or repetition 2) as the second factor...”

Additional concerns on new data and text:

Specific comments:

Figure S2: I do not understand what is plotted on the Y axis? What is a future diabetes incidence? How could the % of future diabetic mice change over time since it represents the % of mouse that will be diabetic in the future. Or is it the % of diabetic mice? If so, please correct the Y axis title everywhere. Also, description of the data from this figure in the main text is long and unnecessary. It could be shortened.

Response: In response to the first part of the question related to the understanding of the figure, each graphic compares the future diabetes incidence of the two groups of mice (low abundance or richness versus high abundance or richness). This diabetes incidence is “future” because at the time points (6, 12, or 20 weeks) when the gut microbiota is analysed (and the low/high abundance/richness of the corresponding taxon is determined), the mice have not yet developed diabetes. However, as we followed the incidence until 40 weeks, we know which will develop the disease in the future and at what age this will happen. The percentage of future diabetic mice can change over the weeks since we know the exact age at which every mouse will develop diabetes. For example, we compare the incidence that coNOD mice with high and low richness at 12 weeks of age will develop in the future (after 12 weeks). The future-diabetic mice of each group will develop the disease at different ages between 14 and 24 weeks.

In response to the second part of the question related to the length of the text describing figure S2, we agree with the reviewer and consequently, we reduced this fragment as follows:

In a subsequent analysis, we selected significant bacterial taxa and Chao1 index and grouped future-diabetic and future-resistant T1D-prone mice (isolated NOD, cohoused NOD, and isolated 116C-NOD) based on high or low relative abundance of corresponding taxa or richness. T1D incidence was compared (Supplementary Fig. 2). Isolated NOD mice at 12 weeks with low Cyanobacteria abundance had a lower future-diabetes incidence than those with high abundance ($p=0.039$). At the same age, cohoused NOD mice with high richness (Chao1 index) showed significantly reduced future-T1D incidence compared to those with low richness ($p=8 \times 10^{-4}$). Additionally, cohoused NOD mice with high Proteobacteria abundance had a lower future-diabetes incidence ($p=0.006$). At six weeks, isolated 116C-NOD mice with low Cyanobacteria YS2 genus and Tenericutes abundance exhibited reduced future-T1D incidence ($p=0.002$ and $p=0.010$, respectively). Moreover, isolated 116C-NOD with low Alphaproteobacteria RF32 genus abundance displayed reduced future-diabetes incidence at 12 weeks ($p=0.012$) and 20 weeks ($p=1 \times 10^{-4}$) compared to those with high abundance. Finally, across all three T1D-prone mice, those with low Clostridium (Ruminococcaceae family) abundance had significantly lower future-diabetes incidence compared to those with high abundance at 20 weeks ($p=1 \times 10^{-4}$).

Line 335 and 338: NOD.Rag-/- mice also lack B cells and 116-C.NOD.Rag-/- mice lack a polyclonal B cell population (therefore B cells that could recognize microbial antigens). B cells could therefore be at cause, especially as the transfer of NOD B cells also decreased the presence of filamentous bacteria (Fig. 7b).

Response: We appreciate the reviewer’s comment and the opportunity to clarify this issue.

In other experiments developed in our lab (manuscript in preparation), we found that 116C B cells can recognize intestinal microbiota bacterial antigens of intestinal microbiota (by means of cross-reactivity). Taking this into account and also: (1) that 116C B cells are responsible for changes in the gut microbiota of 116C-NOD.RAG-2-/- (please see current figures 8d and 9c); and

(2) the fact that the transfer of NOD B cells provides a significant difference compared to the transfer of total NOD spleen, but it is not significantly different from the transfer of NOD.RAG-2^{-/-} spleen; we conclude that the results suggest that T cells are the lymphocytes responsible for the alteration observed in SFB proliferation.

Line 477: Similarly, the authors should write “T cells disorders are a driver” rather than “the driver”.

Response: We acknowledge the reviewer’s comment, and consequently, we replaced “the” with “a” in this sentence.

Line 349: “The presence of lymphocytes is a requirement for Bifidobacterium colonization”. Is it possible that some strains (B6, NOD, etc) are simply colonized in the animal facility while others are not (NOD.Rag^{-/-} etc). To prove that the presence of lymphocytes is a requirement would require co-housing of NOD (or B6) with NOD.Rag^{-/-} mice to see if colonisation can take place. Same thing for the claim made in line 361/362.

Response: We appreciate the reviewer’s comment. Although the reviewer's statement is possible, we believe that the probability of this fact is lower compared to the argument described in lines 386-392. However, we agree that to fully confirm our argument, a cohousing experiment or a fecal transfer experiment could be conducted. Therefore, we corrected the following sentences:

- Lines 386-7: ...the results suggested that the presence of lymphocytes is a requirement for *Bifidobacterium* colonisation...
- Lines 396-7: “...the results suggested that, unlike *Bifidobacterium*, *Allobaculum* did not require T nor B lymphocytes ...”

Line 372: 7a should be 7c

Response: Reference to figure is now corrected to current figure 8c

Reviewer #5 (expertise microbiota and T1D):

The authors have now added new data showing the impact on intestinal permeability and immune populations in the GALT tissues following co-housing with 116C-NOD mice. This new data improves the manuscript by adding extra evidence for how the 116C microbiota may be impacting the host. However, the specific microbiota pathways that are mediating these effects are still unknown and knowledge of this is limited by only using 16S amplicon profiling of the microbiota not metagenomics or metabolomics (e.g. SCFA measurement or more comprehensive metabolite profiling). Therefore, this remains a significant limitation that needs to be acknowledged in the discussion.

Response: Conducting co-housing and cell transfer experiments has provided conclusive evidence of the gut microbiota's direct involvement in the development of T1D. However, pinpointing the precise mechanisms responsible for the fluctuations in T1D incidence would necessitate the formulation of novel experiments. These would encompass the utilisation of advanced techniques such as shotgun DNA/RNA metagenomics or metabolomics approaches, followed by the transplantation of the identified microorganisms or the transfer of microbial

products into an animal model. In response to the reviewer's suggestion and also as recommended by reviewer #4, we are in the process of designing a new project specifically aimed at addressing this question. We have now added this limitation in the discussion section as follows:

“...To gain a deeper understanding of the specific pathways and mechanisms connecting gut microbiota to T1D, advanced techniques of microbiota analysis, such as shotgun DNA/RNA metagenomics or metabolomics approaches, along with direct *ex vivo* cytokine profiling analyses, are necessary...”

4. The lack of direct functional microbiome profiling means it is hard to explain some of the bacterial changes which are both lower in protected coNOD mice but higher in iso116C-NOD mice (*Sutterella* and *Proteobacteria*). Furthermore, these two taxa were not predictive of future diabetes in the new Suppl Fig 2. Therefore the relevance of changes in these taxa is unclear and the paragraph in the discussion (lines 458-466) on these two taxa should be removed.

Response: We have now removed the text related to these taxa in the discussion.

5. Cytokine levels were not checked in vivo – this could have been done by FACS with Intracellular cytokine staining *ex vivo* or via IHC or IF staining on tissue sections. This also should be acknowledged as a limitation in the discussion.

Response: We appreciate the reviewer's comment. We have now added this point to the discussion as follows:

“...To gain a deeper understanding of the specific pathways and mechanisms connecting gut microbiota to T1D, the utilisation of advanced techniques of microbiota analysis such as shotgun DNA/RNA metagenomics or metabolomics approaches, along with direct *ex vivo* cytokine profiling analyses, are necessary...”

6. Lines 514-517: It is noted that *Parabacteroides* is lower in multiple sclerosis patients. Yet, *Parabacteroides* was found to be higher in children that progressed to type 1 diabetes (Stewart et al Nature 2018: Temporal development of the gut microbiome in early childhood from the TEDDY study). *Parabacteroides* was also increased in adults with T1D after consuming a resistant starch diet that is associated with diabetes protection in mice (Bell et al 2022 Microbiome). Therefore, it remains unclear whether *Parabacteroides* has a beneficial or detrimental role in T1D. The discussion should be updated to reflect this.

Response: We acknowledge the reviewer's comment and concur with the necessity to add a reference to this point. We added this information in the discussion section as follows (underlined text):

“...We found that 116C B cells could specifically induce the colonisation of bacteria such as *Adlercreutzia* and *Parabacteroides*. Interestingly, both genera were less abundant in the gut microbiota of multiple sclerosis patients.^{29,31} Parallely, in our work, NOD mice displayed lower levels of these genera compared with C57BL/6J controls. However, in humans, *Parabacteroides* was associated with T1D onset.³⁰...”

We preferred to refer to the study of Stewart et al. since the study of Bell et al. includes the variable of a diet treatment, which could mask the differences between “pure” cases and controls.

Minor point: Lines 199-200 T cell populations are labelled as exhausted and anergic based on surface expression of PD-1 and Lag-3 or CD73 and FR4. Yet more conclusive evidence (e.g. functional or confirming expression of more markers such as EOMES) would be needed to definitively define these populations as such. Therefore, the populations should be referred to as 'exhausted-like' and anergic-like' cells. It would also be helpful for the reader to include the markers gated for the immune populations in the Figure (e.g. in the axis labels or in the legend).

Response: We appreciate the reviewer's comment. Populations previously referred to as "exhausted" (PD-1⁺ and LAG-3⁺) and "anergic" (CD73^{high} FR4^{high}) are now referred to as 'exhausted-like' and anergic-like'. Moreover, we added the markers of each population in the figure legends (please see Figure 4 and Supplementary Figure 5).